# Structure and functional dynamics of the mitochondrial Fe/S cluster synthesis complex

Michal T. Boniecki[1], Sven A. Freibert [2], Ulrich Mühlenhoff[2], Roland Lill [2,3] & Miroslaw Cygler [1,4]

Iron–sulfur (Fe/S) clusters are essential protein cofactors crucial for many cellular functions including DNA maintenance, protein translation, and energy conversion. De novo Fe/S cluster synthesis occurs on the mitochondrial scaffold protein ISCU and requires cysteine desulfurase NFS1, ferredoxin, frataxin, and the small factors ISD11 and ACP (acyl carrier protein). Both the mechanism of Fe/S cluster synthesis and function of ISD11-ACP are poorly understood. Here, we present crystal structures of three different NFS1-ISD11-ACP complexes with and without ISCU, and we use SAXS analyses to define the 3D architecture of the complete mitochondrial Fe/S cluster biosynthetic complex. Our structural and biochemical studies provide mechanistic insights into Fe/S cluster synthesis at the catalytic center defined by the active-site Cys of NFS1 and conserved Cys, Asp, and His residues of ISCU. We assign specific regulatory rather than catalytic roles to ISD11-ACP that link Fe/S cluster synthesis with mitochondrial lipid synthesis and cellular energy status.

[1] Department of Biochemistry, University of Saskatchewan, 107 Wiggins Road, Saskatoon, SK, Canada S7N 5E5. [2] Institut für Zytobiologie und Zytopathologie, Philipps-Universität, Robert-Koch-Strasse 6, 35032 Marburg, Germany. [3] LOEWE Zentrum für Synthetische Mikrobiologie SynMikro, Hans-Meerwein-Strasse, 35043 Marburg, Germany. [4] Department of Biochemistry, McGill University, 3649 Promenade Sir William Osler, Montreal, QC, Canada H3G 0B1. Michal T. Boniecki and Sven A. Freibert contributed equally to this work. Correspondence and requests for materials should be addressed to R.L. (email: Lill@staff.uni-marburg.de) or to M.C. (email: Miroslaw.cygler@usask.ca)

Biogenesis of iron–sulfur (Fe/S) proteins is an ancient, protein-mediated process existing in all three kingdoms of life. Eukaryotes contain the iron–sulfur cluster assembly (ISC) machinery in mitochondria and the cytosolic iron–sulfur protein assembly system[1–3]. The mitochondrial ISC machinery is highly conserved from yeast to man, and consists of 18 known proteins most of which were inherited in evolution from bacteria[4–6]. Defects in mitochondrial Fe/S protein assembly cause severe human diseases with often fatal outcome[7, 8]. Biogenesis involves the *de novo* synthesis of a Fe/S cluster on the scaffold protein ISCU, trafficking of the cluster via transient association to various targeting factors, and finally cluster insertion into specific recipient apoproteins. A critical initial step in Fe/S cluster synthesis is the supply of sulfur by the cysteine desulfurase NFS1. The enzyme belongs to a subfamily of pyridoxal 5′-phosphate (PLP)-dependent transaminases that convert free L-cysteine to alanine and an enzyme-bound persulfide (-SSH) group[9, 10]. The biochemical mechanism of persulfide generation on a conserved cysteine residue was worked out with NifS, the bacterial founding member of this family[11], followed by determination of crystal structures of prokaryotic NifS/IscS[12–14] or SufS desulfurases[15],

and their complex with the scaffold protein IscU[16, 17]. One of the latter structures contains a bridging [2Fe–2S] cluster.

In contrast to the bacterial ISC system, the mitochondrial machinery contains and functionally depends on the small protein ISD11 (also termed LYRM4), a member of the LYRM protein family[18–20]. LYRM proteins form complexes with acyl carrier protein (ACP) that is involved in mitochondrial fatty acid synthesis and lipoic acid formation[21]. LYRM-ACP complexes are associated with, e.g., respiratory complexes I, II, III, and V and their assembly intermediates[22–24]. Recently, yeast Acp1 has been identified as a crucial constituent of the Nfs1-Isd11 complex[25], and it was shown that *E. coli* ACP co-purifies with bacterially expressed NFS1-ISD11[26]. Depletion of either Acp1 or Isd11 leads to the disappearance of the Nfs1-Isd11 complex and to Fe/S protein biogenesis defects[18, 19, 25].

De novo synthesis of a [2Fe–2S] cluster on the ISCU scaffold protein further involves the function of reduced ferredoxin (FDX2), presumably converting the persulfide sulfur ($S^0$) to sulfide ($S^{2-}$)[27, 28], and frataxin (FXN), which may fulfill a regulatory role in persulfide sulfur transfer from NFS1 to ISCU and/or provide iron[29, 30]. To date, neither the precise roles of ISD11 and

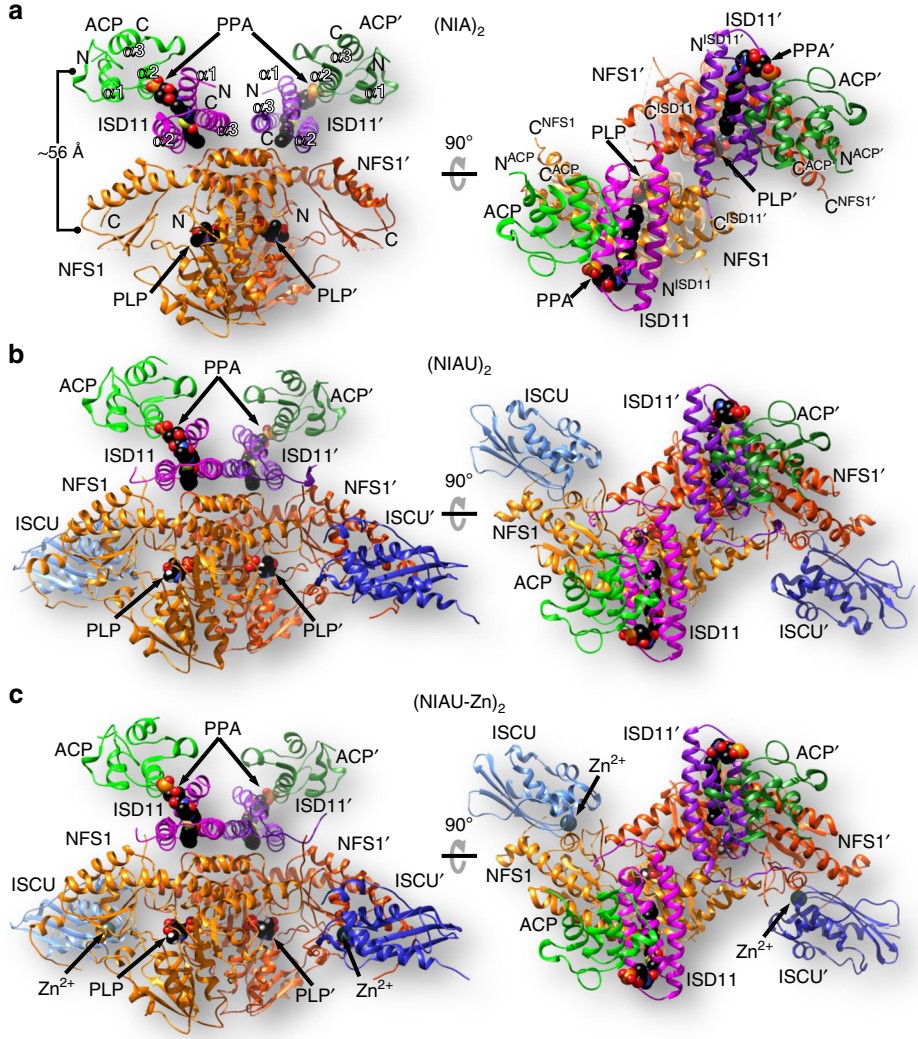

**Fig. 1** Crystal structures of (NFS1-ISD11-ACP)$_2$ and (NFS1-ISD11-ACP-ISCU)$_2$ complexes. Overview of the 3D structures of the **a** (NIA)$_2$, **b** (NIAU)$_2$, and **c** (NIAU-Zn)$_2$ complexes. NFS1 is depicted in orange, ISD11 in magenta or purple, ACP in green, and ISCU in blue. Pyridoxal phosphate (PLP) and the phosphopantetheine with its fatty acyl chain (PPA) are depicted in black as spheres. In part **a** the ~56 Å distance between the centers of NFS1 and ACP as well as the N and C termini are indicated

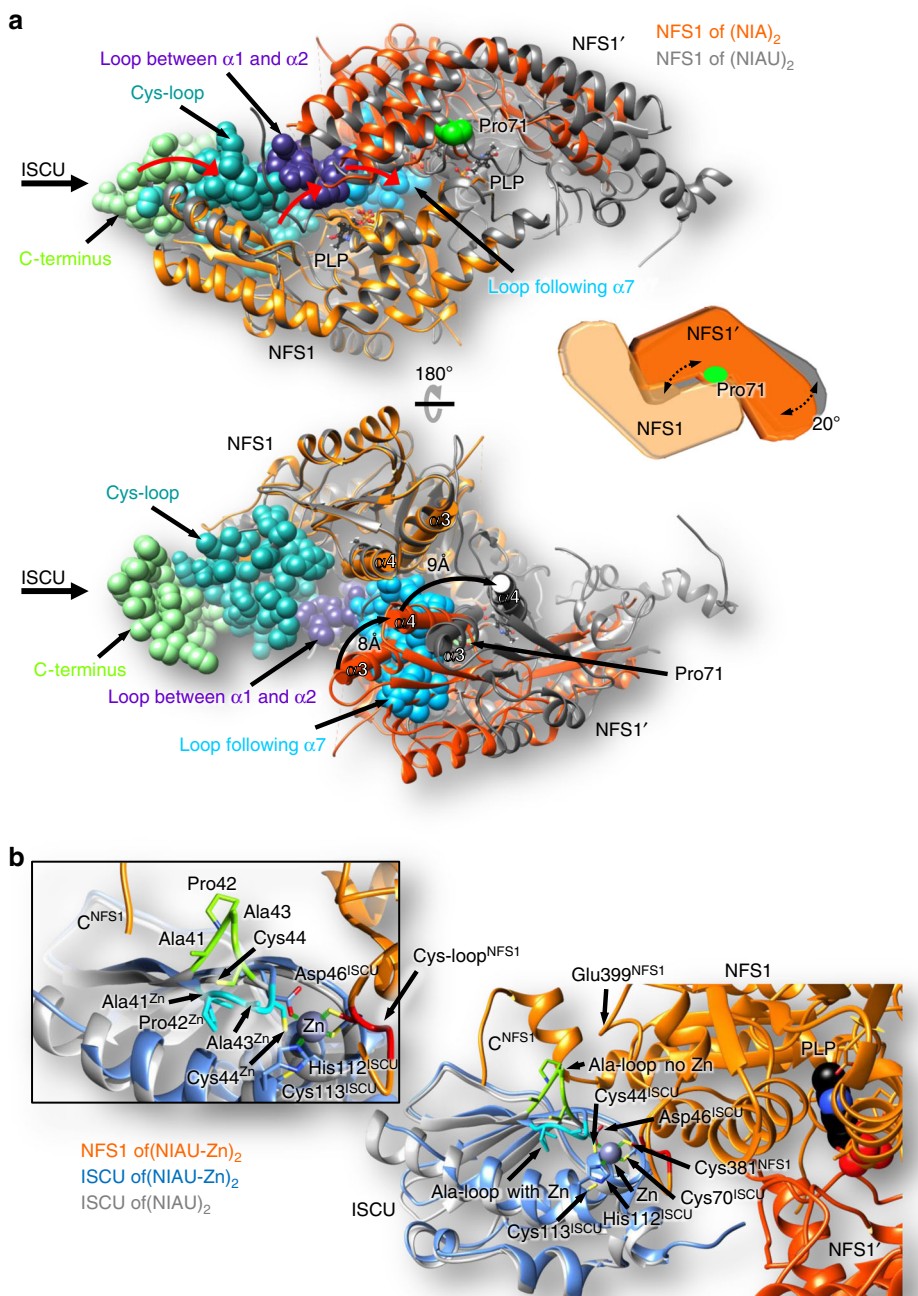

**Fig. 2** Conformational changes of NFS1 upon ISCU binding. **a** The NFS1 dimer rotation visualized by superposition of NFS1 as present in $(NIA)_2$ (orange) or $(NIAU)_2$ (gray). The rotation pivot point is located close to Pro71 (green) in front of the PLP moiety (black). Regions of NFS1 disordered in $(NIA)_2$, yet structured in $(NIAU)_2$ complexes are shown by spheres. They connect the ISCU binding site and helices $\alpha3$ and $\alpha4$ in a "domino-like" fashion (red arrows). The tips of helices $\alpha4$ in the $(NIA)_2$ complex are close together and provide new stabilizing interactions in place of disordered regions. As a result of ordering due to ISCU binding, NFS1 helices $\alpha3$ and $\alpha4$ re-orient by 8 Å and 9 Å, respectively and the 'standard' interface is recreated. The cartoon illustrates the movement of the NFS1 dimer. **b** Mode of ISCU binding to NFS1 in the $(NIAU)_2$ and $(NIAU-Zn)_2$ complexes. Upon Zn binding to the active-site Cys381$^{NFS1}$, the Cys-loop (red) becomes fully structured. Additionally, Zn is coordinated by Asp46, Cys70, and His112 of ISCU. The Ala-loop (Ala41-Cys44) of ISCU (cyan in $(NIAU)_2$ and green in $(NIAU-Zn)_2$) is rearranged upon Zn binding (inset)

ACP in the NFS1-catalyzed reactions (i.e., sulfur release from cysteine, persulfide generation, and sulfur transfer) nor the molecular mechanism of [2Fe–2S] cluster synthesis on ISCU are known. Here, we report crystal and solution structures of human NFS1 in association with ISD11-ACP, ISCU, FDX2, and FXN, together forming the 'core ISC complex'. Our structural and complementary biochemical studies provide important functional

insights into the molecular mechanisms of de novo [2Fe–2S] cluster synthesis on the ISCU scaffold protein and the dynamics of the core ISC complex during this process. We show that eukaryote-specific ISD11 binds to NFS1 distal from the desulfurase active site and does not directly participate in catalysis. ACP binds solely via ISD11 and is not required for Fe/S cluster assembly in vitro. Our studies indicate regulatory roles of the

small factors ISD11-ACP, connecting Fe/S cluster synthesis to mitochondrial fatty acid synthesis and respiratory function, thus potentially allowing the sensing of the energetic status of the cell.

## Results

### Crystal structures of NFS1-ISD11-ACP with or without ISCU.

His-tagged human NFS1 was co-expressed with ISD11 in *E. coli* and purified. Somewhat unexpectedly, the *E. coli* ACP co-purified with NFS1-ISD11. The (NFS1-ISD11-ACP)$_2$ complex (hitherto (NIA)$_2$) was successfully crystallized and its structure determined at 2.75 Å resolution by molecular replacement using *E. coli* IscS[12] and ACP[31] as models (Fig. 1a). The ISD11 model was built into the unassigned electron density. In addition to the (NIA)$_2$ complex, we have purified and crystallized two different four-protein complexes containing also the human scaffold protein ISCU. For crystallographic studies we have used the M107I mutant of ISCU in the expectation that frataxin would be dispensable as part of the core ISC complex[32]. Crystals obtained from purifications in the presence of ethylenediaminetetraacetic acid (EDTA) (termed (NIAU)$_2$) did not contain divalent ions and diffracted to 3.15 Å (Fig. 1b), while those from purifications without EDTA contained a Zn ion bound between NFS1 and ISCU ((NIAU-Zn)$_2$) and diffracted to 3.3 Å (Fig. 1c). The three structures with a central NFS1 dimer identify the binding sites of two interacting ISD11 monomers on the same side of the NFS1 dimer and distal from its catalytic center. They further reveal that the two ACP are solely associated with ISD11 opposite of NFS1. Two ISCU bind to the C-terminal regions of NFS1 (Fig. 1b, c). The NFS1-ISD11-ACP sub-complexes in all three structures are similar and superimpose with a root-mean-square deviation (rmsd) of 0.9 Å for 409 Cα atoms (Supplementary Fig. 1a).

### Structures of NFS1 dimer and NFS1-ISCU sub-complex.

Consistent with the evolutionary origin, the NFS1 structure is similar to that of bacterial PLP-dependent cysteine desulfurases, a subfamily of transaminases (Supplementary Fig. 1b). The large domain of NFS1 (residues 71–315) harbors the PLP cofactor, which is covalently bound to Lys258 as an internal aldimine. The loop containing active-site Cys381 (Cys-loop) is ordered in the (NIAU-Zn)$_2$ complex, with Cys381 acting as a Zn$^{2+}$ ligand together with several ISCU residues (see below), while in the Zn-free (NIAU)$_2$ complex the NFS1 residues Ala380-Leu386 are disordered, as frequently noted in bacterial cysteine desulfurases[12, 14]. The elements of NFS1 discussed here are marked in Supplementary Fig. 1c, d.

In the (NIA)$_2$ complex, some segments of NFS1 are disordered. This includes the C-terminal segment, the Cys-loop, and two loops adjacent to the disordered C-terminal segment on the opposite monomer (Fig. 2a, spheres and Supplementary Fig. 1d). The latter two loops correspond to regions that usually form the 'standard' dimer interface in bacterial desulfurases and transaminases (Supplementary Fig. 1b). As a result of this structural disorder, the dimer assumes a 'non-standard' conformation through a ~ 20° rotation of its monomers towards each other along the direction perpendicular to the long axis of the dimer with the pivot point near Pro71$^{NFS1}$ at the N-terminus of helix α1. This joins the tips of helices α4 from both monomers providing new stabilizing contacts within the NFS1 dimer (Fig. 2a). Conspicuously, ISCU binding to the C-terminal segment of NFS1 rigidifies the disordered regions through a 'domino effect' and re-creates the 'standard' dimer interface. The pivot point is near the NFS1-ISD11 interface assuring little change in this part of the NFS1 dimer, and hardly affecting the conformation of the NFS1-ISD11-ACP sub-complexes (all NIA

sub-complexes superimpose with a rmsd of 0.9 Å, Supplementary Fig. 1a).

Two ISCU monomers are bound at opposite tips of the NFS1 dimer (Fig. 1b, c). The NFS1 residues binding ISCU include the C-terminal Tyr360-Lys371, His403, Arg432-Thr455, and the tip of the Cys-loop. In particular, the C-terminal Met436-Thr455 wrap tightly around ISCU. These residues are disordered in the (NIA)$_2$ complex and only partially visible in prokaryotic IscS-IscU crystals[16, 17]. ISCU attaches to NFS1 with the end that contains potential ligands for Fe/S cluster binding, Cys44, Asp46, Cys70, His112, and Cys113. In (NIAU)$_2$ this ISCU metal binding site is empty; the conserved 'Ala-loop' (Ala41-Pro42-Ala43-Cys44) of ISCU contacts Glu399 and Trp454 side-chains of NFS1, with Cys44$^{ISCU}$ facing away from the Fe/S cluster binding site (Fig. 2b). The Cys-loop of NFS1 extends towards ISCU, but its tip (Cys381-Ser385) is disordered and not visible in the electron density. In (NIAU-Zn)$_2$ a Zn ion occupies the expected Fe/S cluster synthesis site[17], and is tetrahedrally coordinated by the active-site Cys381$^{NFS1}$ and Cys70, His112 and Asp46 of ISCU (Fig. 2b and Supplementary Fig. 2a–c). The Zn site shows ~ 71% (chain D) and ~57% (chain H) occupancy in the two independent ISCU molecules. Importantly, the presence of Zn causes a reorientation of the Ala-loop of ISCU, with Cys44 approaching and facing the Zn ion (Fig. 2b), whereas Cys113 as part of helix α4 maintains its location most distant to the Zn ion. We conclude that both the NFS1 Cys-loop and the ISCU Ala-loop undergo substantial structural adaptation in response to NFS1-ISCU docking. This allows for a structurally ideal positioning of ISCU to receive the persulfide sulfur from NFS1.

### Structure of ISD11 and its interaction with NFS1.

ISD11 folds into a compact antiparallel three-helix bundle followed by an extended C-terminal loop (Ser77-Asn85) (Fig. 3a). The helices create a tunnel that expands into an open canyon (Fig. 3b). The walls of the tunnel/canyon are predominantly hydrophobic. Residue Leu12 of the LYR motif within helix α1 points towards helix α3 and forms the center of a hydrophobic cluster (Met16, Phe40, Val56, Ala59, Leu63, and aliphatic part of Lys60) adding to the stability of the ISD11 fold (Fig. 3a). Tyr13 and Arg14 of LYR face outside and interact with ACP (see below). Each ISD11 contacts both NFS1 monomers (Fig. 1). Most contacts are with one NFS1 molecule through the sides of helices α2 and α3, in part engaging polar-ionic interactions (Arg68$^{ISD11}$–Asp75$^{NFS1}$ and Arg35$^{ISD11}$–Glu314$^{NFS1}$) and hydrogen bonds (Tyr31$^{ISD11}$–Arg72$^{NFS1}$ and Asp38$^{ISD11}$–Tyr317$^{NFS1}$) (Fig. 3c, bottom left). That these interactions are important is evident from the fact that mutations of R68L$^{ISD11}$ and R72Q$^{NFS1}$ are associated with human Fe/S deficiency diseases[33, 34]. There are additional, mostly hydrophobic interactions of the C-terminus of ISD11 (Thr78-Ile83) with the N-terminus of NFS1 (Arg57-Leu59) (Fig. 3c, bottom middle). The NFS1 interface includes predominantly residues from three α-helices (α1, α8, α11; Supplementary Fig. 1c, d) on top of NFS1 facing ISD11, as well as the N-terminus of NFS1. At the center of the NFS1 interface is Phe413 forming herring bone-type interactions with Tyr28$^{ISD11}$ and Tyr31$^{ISD11}$ (Fig. 3c, bottom right). ISD11 contacts the second NFS1 through the C-terminus of helix α3 and through Tyr76 and Thr78 (Fig. 3c, bottom middle).

The interaction site of ISD11 on the surface of the NFS1 dimer includes a hydrophobic patch centered on the N-terminal NFS1 α1-helices Leu69-Ile82 (Supplementary Fig. 3a, b). These hydrophobic residues could contribute to the observed aggregation of NFS1 when expressed alone. In the NFS1-ISD11 complex, this hydrophobic patch is covered by the C-terminal helices α3 of both ISD11 monomers, thus suppressing NFS1 aggregation.

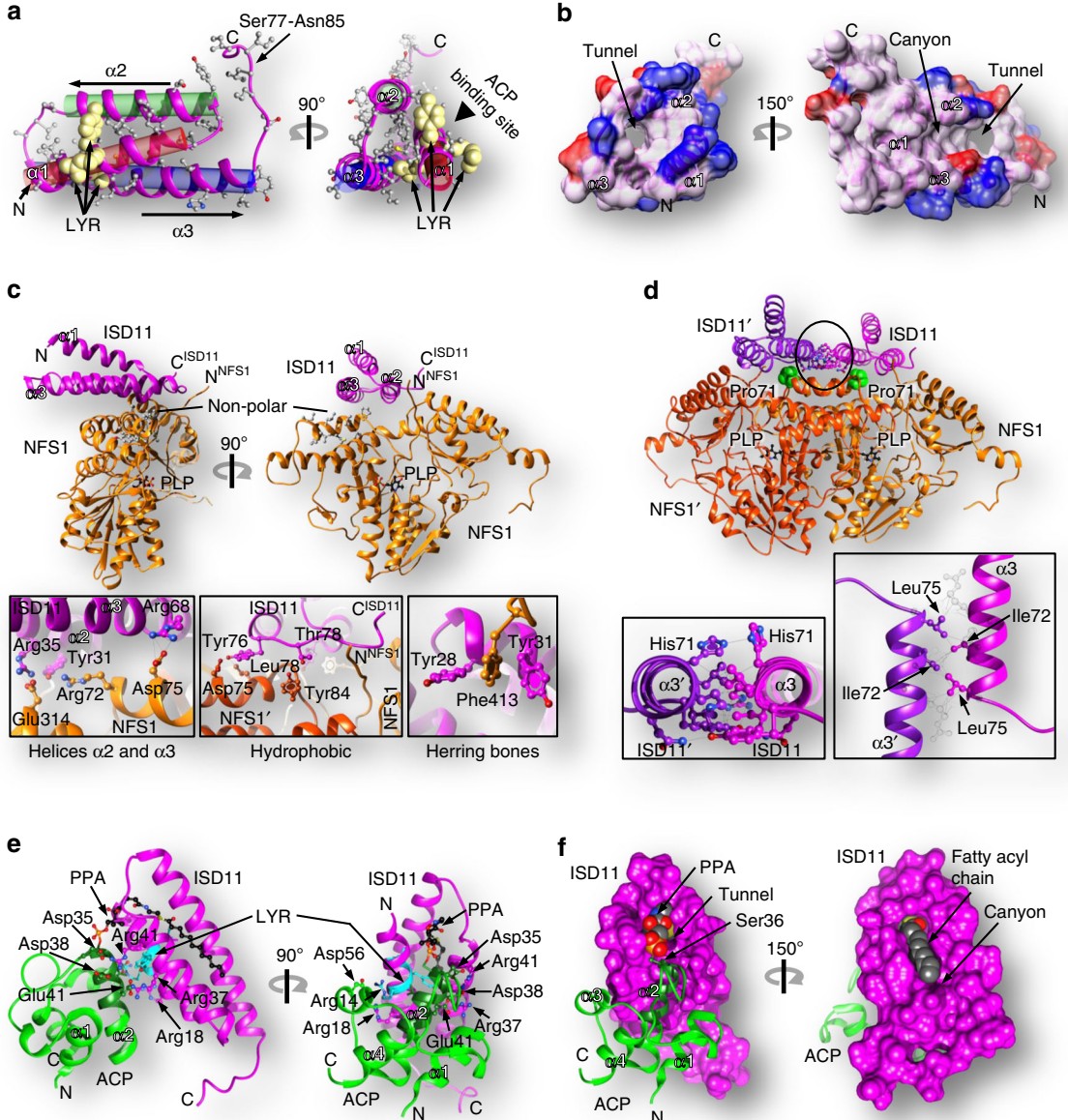

**Fig. 3** Structure of ISD11 alone and in complex with NFS1 or ACP. Structures of subunits and sub-complexes were taken from the (NIAU)$_2$ complex. **a** Structure of ISD11. All hydrophobic residues found in ISD11 are in gray. Helices α1 (red), α2 (green), and α3 (blue) are antiparallel (α2 and α3) or tilted by ~21° (α1). N and C termini are marked. The LYR motif is shown in khaki by spheres. **b** Surface representation of ISD11 shows a hydrophobic tunnel between the three α-helices and a canyon. Surface is colored in light-magenta except for positively and negatively charged residues in blue and red, respectively. **c** Hydrophobic and polar interactions between ISD11 (magenta) and NFS1 (orange). Boxes show details of the interaction interfaces. **d** Interaction of the ISD11 and NFS1 dimers. Insets show details of ISD11–ISD11 interaction. **e** Structure of the ISD11-ACP sub-complex. Phosphopantetheine with a fatty acyl chain (PPA) is colored in black, and the LYR motif in cyan. **f** Surface representation of ISD11 in the ISD11-ACP sub-complex shows that PPA (red balls) and part of the fatty acyl chain (gray balls) bind to a hydrophobic canyon of ISD11, while the rest of the fatty acyl chain penetrates the hydrophobic tunnel of ISD11

There is no such continuous hydrophobic surface in bacterial IscS-like proteins, which instead contain one or more charged residues in this region (e.g., negatively charged Asp30 in *E. coli* IscS) (Supplementary Fig. 3a). The two ISD11 molecules within the complex contact each other through the sides of the C-terminal segments of helices α3 (Fig. 3d). These contacts are of predominantly hydrophobic nature and involve the side-chains of Ile72, Leu75, but also several intermolecular hydrogen bonds (Fig. 3d, bottom parts). These ISD11–ISD11 interactions contribute to the stability of all complexes, since, being close to the pivot point of the NFS1 dimer rearrangement, they are only marginally affected by these conformational changes (Fig. 3d).

Overall, the architecture of ISD11-NFS1 association structurally explains the stabilizing role attributed to eukaryote-specific ISD11 in biochemical studies[18, 19]. This suggests a regulatory rather than catalytic function of ISD11 (see below).

**Structure of ACP and its interaction with ISD11**. *E. coli* ACP binds solely to ISD11. The structure of ACP in the complex is nearly identical to that of apo-ACP (PDB code 2FAE[31]; Supplementary Fig. 3c). ACP contains a four-helix bundle with long connecting loops. It contacts ISD11 via helix α2 (Asp35-Met44) and Asp56 from the loop following this helix. The interface on

ISD11 involves helices α1 and α2 (Fig. 3e). The contacts are mostly through hydrophilic and charged side-chains (salt bridges), including the hydrogen-bonded Tyr13[ISD11] of the LYR motif. A major contribution to binding comes from the ACP's Ser36-linked phosphopantetheine and its attached fatty acyl chain that flipped out of the pocket in ACP and entered the

hydrophobic tunnel of ISD11 described above (Fig. 3b, f). The electron density allowed us to place a fatty acyl chain with 12 carbon atoms (Supplementary Fig. 3d). The molecular mass of ACP determined by mass spectrometry of dissolved crystals was 9073 Da, which corresponded well to ACP with a 14 carbon fatty acyl chain (calculated 9072 Da) and indicated that the last two

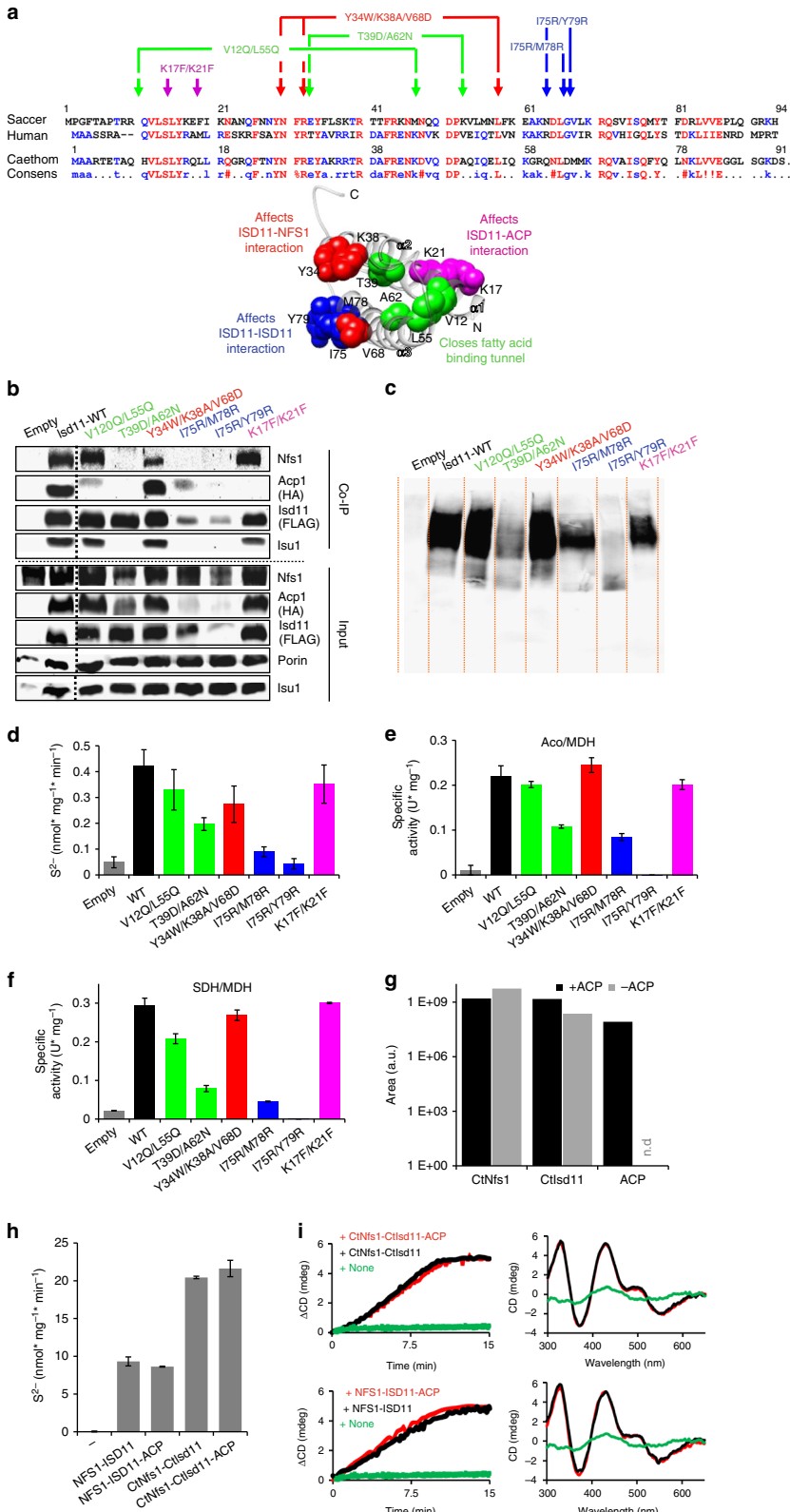

carbons were disordered. Together, the mode of ACP binding to NFS1-ISD11 renders a direct role of ACP in catalysis unlikely and rather suggests a regulatory function (see below).

**Mutagenesis and biochemistry of ISD11-NFS1-ACP interactions.** To correlate the mode of ISD11 and ACP binding in the $(NIA)_2$ and $(NIAU)_2$ crystal structures (Fig. 1) with the in vivo situation, we employed yeast genetics and introduced site-specific mutations into *Saccharomyces cerevisiae ISD11*, which were predicted from the structures to specifically affect ISD11 binding to either NFS1 or ACP (Fig. 4a). We employed Gal-ISD11 yeast cells, in which the endogenous wild-type copy of Isd11 can be depleted by growth in the presence of glucose[19]. This causes a severe growth retardation and Fe/S protein biogenesis defects that can be reversed by expression of FLAG-tagged wild-type Isd11 (Fig. 4 and Supplementary Fig. 4). To test the effects of various *ISD11* mutations, mutant Isd11-FLAG proteins were expressed from a plasmid in Isd11-depleted Gal-ISD11 cells harboring HA-tagged Acp1[25]. Cells were analyzed for effects on growth, and isolated mitochondria were tested for Isd11 interaction with Nfs1, Acp1, and Isu1, cysteine desulfurase activity, and the functional consequences on Fe/S protein biogenesis.

Isd11 mutations I75R/M78R (human I72/L75) and I75R/Y79R (human I72/Y76) were predicted to affect the Isd11–Isd11 interaction and stability, and may impact Isd11 binding to Nfs1 (Fig. 4a; blue). Indeed, both Isd11 mutant proteins did not restore growth of depleted Gal-ISD11 cells (Supplementary Fig. 4a, b). Immunostaining of isolated mitochondria showed strongly reduced amounts of Isd11, Acp1 and, in case of the I75R/Y79R mutation, also of Nfs1 (Fig. 4b, input). In contrast, levels of Isu1 and the mitochondrial control protein porin were unchanged. Apparently, these Isd11 mutations affect the stability of all Nfs1-Isd11-Acp1 sub-complex constituents. We then performed co-immunoprecipitation (Co-IP) experiments using (mutant) Isd11-FLAG as a bait. Nfs1, Acp1, and Isu1 but not porin were co-immunoprecipitated with wild-type Isd11-FLAG, yet not with the Isd11 variants I75R/M78R and I75R/Y79R (Fig. 4b, co-IP). A severe functional impairment of these two Isd11 mutant proteins was also suggested by the strongly diminished amount of the Nfs1-Isd11 high-molecular mass complex detected by blue-native PAGE[18, 19], and a low cysteine desulfurase activity in isolated mitochondria (Fig. 4c, d). Further, we observed a severe drop in the enzyme activities of the mitochondrial Fe/S proteins aconitase and succinate dehydrogenase (SDH; Fig. 4e, f; Supplementary Fig. 4c–e). Together, these results show the in vivo importance of the conserved yeast Isd11 residues I75-M78-Y79 (human I72-L75-Y76) for Isd11 interaction with the ISC

biosynthetic complex, which then may cause instability of Isd11 and, in turn, of the entire complex.

Isd11-Y34W/K38A/V68D (human Y31/R35/V65) was predicted to impair Isd11-Nfs1 interactions (Fig. 4a; red). Isd11-depleted Gal-ISD11 cells expressing this protein supported wild-type growth (Supplementary Fig. 4a, b), contained almost normal levels of Nfs1-Isd11 complex and desulfurase activity, and maintained Fe/S protein biogenesis activity (Fig. 4b–f; Supplementary Fig. 4c–e). Nevertheless, a lower Co-IP efficiency of Isd11-Y34W/K38A/V68D with Nfs1, and in turn Isu1, was observed, while the association with Acp1 was unchanged (Fig. 4b, co-IP). This shows the specific importance of these Isd11 residues for Nfs1 interaction.

Exchanges V12Q/L55Q (human V9/I52) and T39D/A62N (human I36/A59) alter the Isd11 hydrophobic channel that accommodates the fatty acyl chain of Acp1 (Fig. 3b). K17F/K21F (human R14/R18) mutation modifies the ACP-interacting residue of the LYR motif (Fig. 4a; magenta and green). All these mutants supported normal growth of depleted Gal-ISD11 cells (Supplementary Fig. 4a, b). Moreover, the Isd11-K17F/K21F and Isd11-V12Q/L55Q mutant proteins showed close to wild-type levels of Nfs1-Isd11 complex, as well as normal desulfurase activity, and Fe/S protein biogenesis (Fig. 4c–f; Supplementary Fig. 4c–e). However, the Isd11 interaction with Acp1 was specifically compromised as seen in Co-IP experiments, while Isd11 association with both Nfs1 and Isu1 remained unaffected (Fig. 4b; Co-IP). Isd11-T39D/A62N showed a stronger phenotype with intermediate levels of Nfs1-Isd11 complex, desulfurase activity, and Fe/S protein biogenesis. This mutation obviously disturbed the stability of the ISC biosynthetic complex because none of its constituents was co-isolated with Isd11-T39D/A62N. Collectively, our mutagenesis results strongly support the ISD11 protein contacts observed in the crystal structures of the ISC complexes and provide important functional insights.

The peripheral attachment of ACP within the $(NIA)_2$ complex raised the question of whether it plays a direct role in Fe/S cluster biogenesis. We addressed this point in vitro for both human NFS1-ISD11 and *Chaetomium thermophilum* CtNfs1-CtIsd11 expressed in *E. coli* and purified with and without bound ACP. We took advantage of the observation that the ACP content of the recombinant NFS1-ISD11 complexes was dependent on the cultivation conditions. When cells were grown in energy-rich medium, a maximum of ACP was associated with purified NFS1-ISD11, while hardly any ACP was recovered after purification from starving cells as judged from mass spectrometric analysis of the soluble protein complexes (Fig. 4g). We used these ACP-containing or ACP-free NSF1-ISD11 preparations to measure desulfurase activity and in vitro Fe/S cluster synthesis by a

**Fig. 4** Biochemical verification of the (NFS1-ISD11-ACP-ISCU)$_2$ structure in yeast. **a** Sequence alignment of ISD11 from *S. cerevisiae* (Saccer), *H. sapiens* (Human), and *C. thermophilum* (Caethom). Mutations are indicated by arrows. Locations of the mutations are indicated in the crystal structure of human ISD11 using the corresponding Saccer nomenclature. **b–f** Gal-ISD11 yeast cells expressing *S. cerevisiae* Acp1-HA were transformed with plasmids encoding either no protein (empty) or wild-type (WT) or the indicated FLAG-tagged Isd11 mutant proteins. Gal-ISD11 yeast cells were depleted of endogenous Isd11 on glucose medium for 40 h, and mitochondrial extracts were prepared. **b** Anti-FLAG co-immunoprecipitation. The immunoprecipitate (Co-IP) and an aliquot of the mitochondrial extract (input) were analyzed by immunostaining for the indicated ISC proteins and porin as a loading control using antibodies against the respective protein or tag (anti-FLAG: Sigma and anti-HA: Santa Cruz). **c** Blue-native gel electrophoresis (7.5–20% polyacrylamide). Immunostaining was against FLAG of Isd11-FLAG. **d** Cysteine desulfurase activity of 200 μg mitochondrial extracts following the in vitro sulfide production. **e**, **f** Enzyme activity assays of mitochondrial aconitase **e** and SDH **f** normalized to malate dehydrogenase (MDH). **g** Orbitrap mass spectrometry of trypsin-digested purified CtNfs1-CtIsd11 solution with and without bound ACP. For quantification the peak area was used. Black and gray bars represent the samples purified from cells growing in high or low energy media, respectively. No ACP could be detected in the latter (n.d.). **h** Desulfurase activity was measured with purified human or *C. thermophilum* NFS1-ISD11 with or without co-purified ACP. The error bars indicate the SD (n ≥ 3). **i** In vitro enzymatic Fe/S cluster synthesis on ISCU2/CtIsu1[28] using purified human or *C. thermophilum* (Ct) ISC proteins. NFS1-ISD11 with or without co-purified ACP as indicated were mixed anaerobically with ISCU, FXN, FDX2, and FdxR, and Fe/S cluster synthesis on ISCU2 was followed by circular dichroism monitoring ΔCD at 431 nm (left). After 20 min CD spectra were recorded to document successful reconstitution of ISCU2 (right). Control reactions were performed without NFS1-ISD11

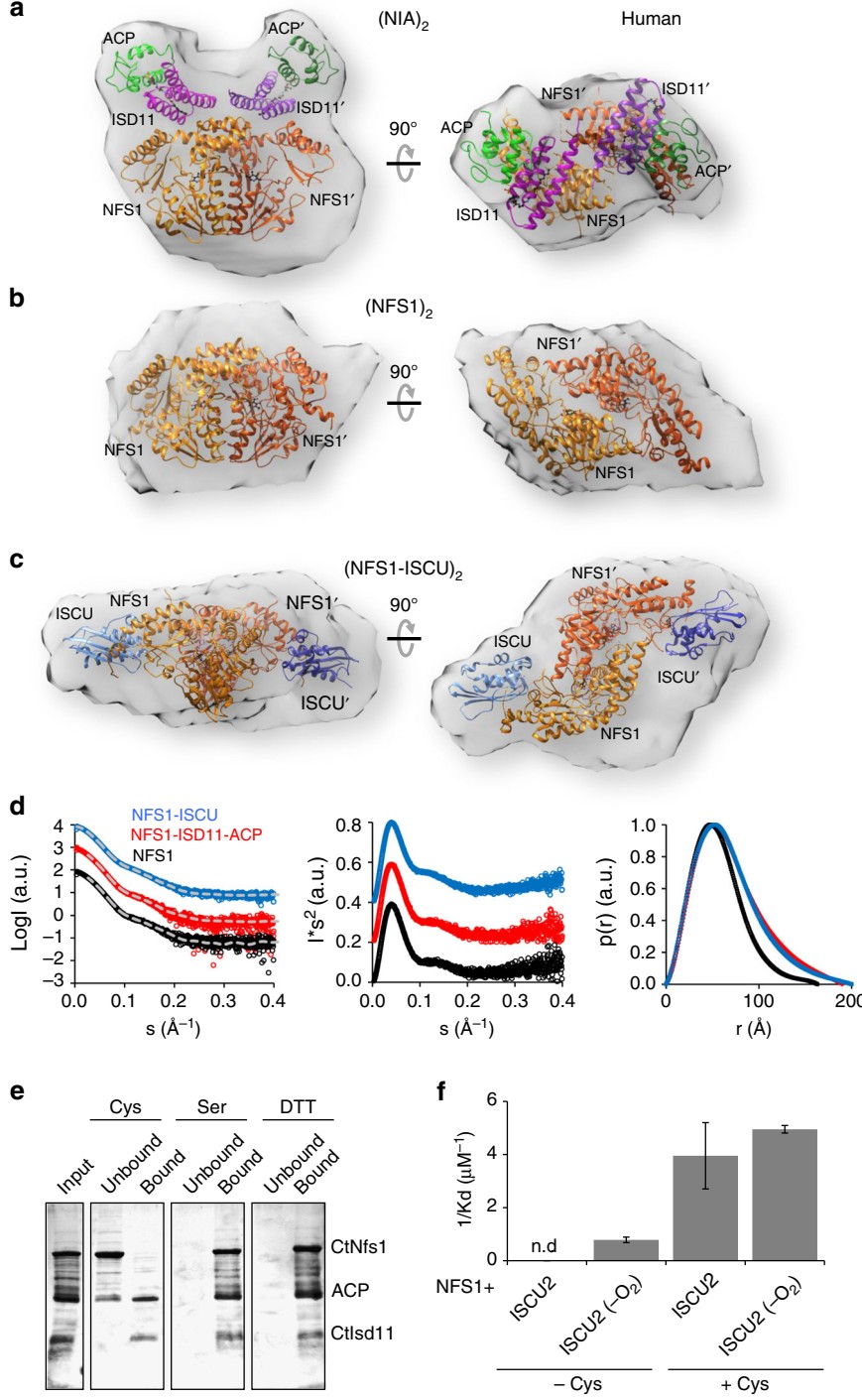

**Fig. 5** SAXS shapes and stability of human ISC complexes. **a–c**, Small angle X-ray scattering (SAXS) analyses were performed with the indicated ISC complexes. Crystal structures of the (NIA)$_2$ complex **a**, the NFS1 dimer part alone **b** or the NFS1-ISCU part of (NIAU)$_2$**c** were fitted into the determined SAXS densities. **d** Scattering curves with respective fits in gray (left) and Kratky plots (middle) for above complexes show no signs of aggregation and suggest a homogeneous solution of the respective complexes. The pair distribution plots (right) show the respective D$_{max}$ values. **e** Cysteine induces dissociation of CtNfs1 from CtIsd11-ACP. The CtNfs1-CtIsd11-His$_6$-ACP complex was incubated with cysteine (Cys), serine (Ser) or dithiothreitol (DTT) under anaerobic conditions and bound to Ni-NTA resin. Proteins eluting from the column (unbound) and resin-bound material (after elution with imidazole) was analyzed by PAGE and Coomassie staining. **f** Equilibrium constants for human NFS1 interaction with ISCU2 were measured by microscale thermophoresis in the absence (left) or presence (right) of cysteine. In part f ISCU was used either aerobically or anaerobically (-O$_2$). The error bars indicate the SD of the sigmoidal fit (see Supplementary Fig. 5e for original data) of three biological replicates

recently described circular dichroism (CD)-based reconstitution assay with purified ISC components[6, 28]. No effects of ACP on desulfurase activity or Fe/S cluster synthesis rates and efficiencies were observed for both human and *C. thermophilum* ISC proteins (Fig. 4h, i). These in vitro data fit to the observed peripheral location of ACP within the complex and suggest that this protein plays no direct mechanistic function in Fe/S protein biogenesis, and rather may execute a regulatory role.

**Solution structures of NFS1-ISD11-ACP-ISCU sub-complexes.** We finally sought to investigate the structures of the various ISC complexes in solution and compare them to the crystal structures. To this end, purified human and *C. thermophilum* ISC components were analyzed by small angle X-ray scattering (SAXS)[35]. Similar envelopes were obtained for the human and *C. thermophilum* orthologs (Fig. 5a, b; Supplementary Fig. 5a, b and Supplementary Tables 1 and 2). First, we analyzed the (NIA)$_2$ complexes (Fig. 5a, d; Supplementary Fig. 5a, c). Since the NFS1 dimer conformations differed in the (NIA)$_2$ and (NIAU)$_2$ crystals, the (NIA)$_2$ SAXS data were compared to both of these conformations. The SAXS data fitted the (NIA)$_2$ crystal structure much better than that of the (NIA)$_2$ sub-complex taken from the (NIAU)$_2$ crystal ($\chi^2 = 0.95$ vs. 23.4; Supplementary Fig. 5d and Supplementary Tables 1 and 2) thus supporting the presence of the (NIA)$_2$ crystal conformation in solution and excluding effects of crystal contacts.

We unexpectedly discovered in our SAXS analyses that incubation of the human or *C. thermophilum* (NIA)$_2$ complexes with cysteine resulted in the dissociation of ISD11-ACP from the NFS1 dimer. The resulting NFS1$_2$ dimer also showed the conformation observed in the (NIA)$_2$ rather than (NIAU)$_2$ crystal structure ($\chi^2 = 4.6$ vs. 72.6, respectively) (Fig. 5b, d; Supplementary Fig. 5b, c and Supplementary Tables 1 and 2). Next, we analyzed by SAXS which NFS1 conformation was present in the human (NFS1-ISCU)$_2$ complex in solution. The human (NFS1-ISCU)$_2$ sub-complex taken from the (NIAU)$_2$ structure showed a good fit to the SAXS data ($\chi^2 = 6.9$), while the (NU)$_2$ model created from the (NIA)$_2$ complex fitted poorly ($\chi^2 = 45.1$) (Fig. 5c, d). We conclude that the conformational change observed in the crystals upon ISCU binding is also occurring in solution.

The dissociative effect of cysteine on the *C. thermophilum* (NIA)$_2$ complex was investigated biochemically by incubating the complex with cysteine and, as a control, serine or the reductant DTT, and subsequent binding to a Ni-NTA affinity resin *via*

**Fig. 6** SAXS shapes of various *C. thermophilum* ISC complexes. **a–c** CtNfs1 (orange) was incubated with CtIsu1 (blue), CtFdx2 (red) or both in 1:1 or 1:1:1 ratios. The NFS1-ISCU crystal structure was fitted to the obtained density **a**. *Ab initio* fitting of FDX2 together with NFS1 into the density resulted in a distinct complex **b**. Both complexes were combined to be fit into the density obtained for the hetero-hexameric complex **c**. **d, e**, Upon addition of CtYfh1 (slate gray) to the CtNfs1-CtIsu1 or CtNfs1-CtFdx2 complexes in a 1:1:1 ratio, complex formation could be observed from the respective SAXS density. The NFS1-ISCU crystal structure was used for ab initio fitting of the positions of NFS1, ISCU, and FXN in the obtained density **d**. To fit NFS1, FDX2, and FXN ab initio into the SAXS density, results from **b** and **d** were combined **e**. **f** Mixing of CtNfs1-CtIsu1-CtFdx2-CtYfh1 yielded an octameric complex. Ab initio fitting into the SAXS densities made use of the above complexes. **g** Scattering curves with respective fits in gray (left), Kratky plots (middle), and pair distribution plots (right) as in Fig. 5. **h** Model of the entire ISC complex based on the crystal and SAXS data. The inset shows the active sites of NFS1, ISCU, FDX2, and FXN. PLP of NFS1 is shown in black, oxygen atoms of negatively charged residues on the iron-binding helix α1 of FXN as red balls, functionally important cysteine sulfurs of NFS1 and ISCU as yellow spheres, and the [2Fe–2S] cluster of FDX2 as spheres. **i** Diagram showing the conformational rearrangement of the NFS1 dimer upon binding of ISCU to a 'standard' cysteine desulfurase dimer conformation

His$_{10}$-tagged CtIsd11. While neither serine nor DTT was able to elute any of the (NIA)$_2$ complex constituents from the resin, cysteine efficiently released CtNfs1, yet CtIsd11 and the majority of ACP remained bound (Fig. 5e). We also noted by equilibrium titrations with microscale thermophoresis (MST) that cysteine increased the binding efficiency of human ISCU2 to NFS1 nine-fold ($K_d = 0.2 \mu M$ with cysteine and $1.7 \mu M$ without; Fig. 5f; Supplementary Fig. 5e). This effect was not due to reduction of disulfide bonds, because neither DTT nor anaerobic conditions affected this affinity increase. These results indicate dynamic conformational changes of NFS1 upon the addition of cysteine causing ISD11-ACP dissociation and tighter ISCU binding.

**Solution structures of NFS1-ISCU-FDX2-FXN sub-complexes**. We further utilized SAXS to get insights into the 3D structure of the complete ISC biosynthetic complex[28]. First, we assembled in solution the complexes containing *C. thermophilum* CtNfs1 plus monomeric CtIsu1 and/or CtFdx2 (Fig. 6a–c, g; Supplementary Fig. 6a and Supplementary Table 1). Since only the reduced form of CtFdx2 efficiently interacts with CtIsu1[28], CtFdx2 was treated with sodium dithionite. SAXS analyses revealed the expected 1:1 and 1:1:1 stoichiometries for the two- and three-protein complexes, respectively. For the CtFdx2-containing complexes we created models using the NMR solution structure of reduced yeast ferredoxin (PDB code 2MJE)[28]. The calculations took into account the known interaction surface between ferredoxin and ISCU proteins[28], and ferredoxin was placed accordingly (Methods section). The SAXS data fitted the models of the respective complexes with good $\chi^2$ values (Supplementary Table 1). Similar results were obtained for human (NFS1-FDX2)$_2$ ($\chi^2 = 4.0$) (Supplementary Fig. 6b, c and Supplementary Table 2). Collectively, the SAXS data suggest that FDX2 binds to NFS1 close to two loops between NFS1 helix α7 and strand β7 as well as helix α8 and strand β9.

Complexes of CtNfs1, CtIsu1, and monomeric CtYfh1 (frataxin) led to the formation of a hetero-hexameric complex (Fig. 6d, g; Supplementary Fig. 6a and Supplementary Table 1). The calculated scattering amplitudes annealed to the experimental data with a $\chi^2$ value of 9.7. While CtIsu1 was placed in the same position as in the complexes above, the calculated location of CtYfh1 on CtNfs1 and the subsequent fitting into the SAXS-derived densities made use of the proposed location of the prokaryotic FXN homolog CyaY[36]. CtYfh1 was located close to CtNfs1 helix α2 and thus snugly fits in a cavity on the large domain of CtNfs1. Interestingly, CtYfh1 did not only interact with the CtNfs1 monomer that exposes its active Cys-loop towards the CtYfh1 neighbors CtIsu1 and CtFdx2, but also with the other CtNfs1 subunit[16]. The simultaneous interaction of CtYfh1 with the two CtNfs1 monomers may be important for the desulfurase activity as a dimer and for the proposed allosteric function of frataxin[37]. The SAXS analysis of the sample containing CtNfs1, CtYfh1, and CtFdx2 revealed a hetero-hexameric complex (Fig. 6e, g, and Supplementary Table 1). The positions of both CtYfh1 and CtFdx2 on CtNfs1 were modeled according to the complexes presented above and yielded a $\chi^2$ value of 10.8. We therefore conclude that CtYfh1 and CtFdx2 bind simultaneously to distinct sites of CtNfs1 in solution.

Finally, we obtained SAXS data for the complete ISC biosynthetic complex in solution. The hetero-octameric complex (CtNfs1-CtFdx2-CtIsu1-CtYfh1)$_2$ was assembled as above and yielded a good $\chi^2$ value of 8.5 (Fig. 6f, g and Supplementary Table 1). CtFdx2, CtIsu1, and CtYfh1 were arranged in a crescent at the tip of a CtNfs1 monomer. The interactions between CtFdx2 and CtIsu1 as well as CtIsu1 and CtYfh1 in this model are consistent with earlier biochemical interaction studies[28, 38, 39].

Interestingly, both the electron-donating [2Fe–2S] cluster of CtFdx2 and the iron-binding helix α1 of CtYfh1[30] were located close to the Fe/S cluster biosynthetic center delineated by the Zn ion present in the (NIAU-Zn)$_2$ complex (Fig. 6h). The overall structure of this 'core ISC complex' provides an initial view how the various ISC components may cooperate during *de novo* Fe/S cluster synthesis.

## Discussion

The crystal and solution structures together with the biochemical investigations presented in this study provide fundamental insights into the global architecture and dynamic function of the mitochondrial Fe/S cluster biosynthesis complex with its central NFS1 desulfurase dimer (Fig. 6h). In this 'core ISC complex', two structurally distant Fe/S cluster biosynthetic centers are formed at opposite tips of the NFS1 monomers by attaching one molecule each of FDX2, ISCU, and FXN. These four ISC components tightly interact with each other to form the catalytic center for *de novo* Fe/S cluster synthesis. The simultaneous and stoichiometric requirement of these ISC proteins for Fe/S cluster synthesis was recently demonstrated in an in vitro reconstitution perfectly supporting the current structural findings[6, 28]. This behavior is strikingly different from the bacterial complex, where CyaY (FXN homolog) and Fdx (FDX2 homolog) were proposed to bind to overlapping positions on IscS[40]. Our study further reveals the positioning of the ISD11-ACP heterodimers that are not present in the bacterial ISC complex. These proteins bind at the NFS1 dimer interface distant from both the Fe/S cluster assembly and desulfurase active centers, yet can be dissociated in response to cysteine addition. Additional dynamic alterations within the core ISC complex were evident upon attachment of ISCU showing its high flexibility during the catalytic cycle. The 3D architectures of the partially assembled and complete core ISC complex represent excellent resources and a prerequisite for the forthcoming structure-based mechanistic dissection of Fe/S cluster assembly in mitochondria.

The catalytic site for Fe/S cluster synthesis is best defined by the (NIAU-Zn)$_2$ crystal structure showing a zinc bound to the persulfide-carrier Cys381 of the swung-out NFS1 Cys-loop and to putative ligands for de novo [2Fe–2S] cluster synthesis on the ISCU scaffold protein. Zn has repeatedly been observed in Fe/S cluster binding sites (e.g., ref.[41]). Our solution structures locate this active center close to the [2Fe–2S] cluster of FDX2 acting as an electron–donor and to helix α1 of FXN serving as a potential iron donor (Fig. 6h). The Ala-loop of ISCU including conserved Cys44 is facing FDX2 and FXN. While it is found in a closed conformation in the presence of Zn in the Fe/S cluster assembly center, it is in an open conformation in the absence of this metal ion, potentially providing access to the assembly center for iron and electrons. The Fe/S cluster synthesis region of ISCU shows a high conformational flexibility in the various ISCU structures. However, equivalent cysteine, aspartate, and histidine residues surround a [2Fe–2S] cluster in the archaeal *Archaeoglobus fulgidus* IscS-IscU complex[17] and Zn in our (NIAU-Zn)$_2$ structure (Supplementary Fig. 2a). Of these, Cys44 and Cys70 are closest to the approaching NFS1 Cys381, while Cys113, biochemically proposed as a persulfide acceptor from NFS1[29], is most distant and part of a fixed α-helix.

In contrast to bacterial desulfurase dimers, the mitochondrial NFS1 dimer shows two strikingly different conformations in the ISCU-deficient and ISCU-containing structures (Fig. 6i, Supplementary Movie 1) indicating flexibility of the NFS1 dimer not observed in bacterial cysteine desulfurases. In the absence of the ISCU scaffold, the C-terminal segment of NFS1 is partially disordered and creates increased flexibility of the residues involved

in the 'standard' dimer interface leading to a rearrangement of the dimer. Binding of the ISCU scaffold rigidifies the C-terminus of NFS1 and restores the 'standard' dimer interface and a dimer conformation common to all other structurally resolved desulfurases[12, 15–17, 42–44]. Both NFS1 conformations present in the crystal structures were confirmed by SAXS analyses in solution suggesting a biochemical relevance of these transitions. Importantly, these conformational changes involve the Cys loop of NFS1, thereby exposing its sulfur-transferring Cys381 to the ISCU residues involved in Fe/S cluster formation. We conclude that ISCU binding dynamically changes the NFS1 structure thereby facilitating sulfur transfer from NFS1.

Our SAXS and biochemical studies revealed a further, so far unknown dynamic behavior of the core ISC complex. Addition of cysteine specifically induced the dissociation of ISD11-ACP from the NFS1 dimer underlining the importance of conformational changes during the catalytic action of the core ISC complex. The effect of cysteine may reflect the activation of NFS1 by persulfide generation on Cys381 at the internal PLP site[45], and/or the outward movement of Cys361-SSH to the active site of Fe/S cluster synthesis. These dynamic alterations induce at least a transient dissociation or loosening of ISD11-ACP from NFS1, and provide biochemical evidence for a regulatory rather than catalytic role of ISD-ACP. This raises the central question of what the functions of eukaryote-specific ISD11 and ACP may be in the mitochondrial ISC complex. Previous results and our new data show that both ISD11 and ACP are not essential for cysteine desulfuration by NFS1, yet both proteins stabilize the NFS1 high molecular mass complex in vivo and prevent NFS1 aggregation[19, 25, 46–48]. These findings are now rationalized by our structural and mutational data showing that ISD11 associates far from both the NFS1-bound PLP and the Fe/S cluster synthesis sites (Fig. 6h). Moreover, binding of the ISD11 dimer shields a hydrophobic patch at the NFS1 dimer interface. These features indicate a regulatory rather than direct catalytic role of ISD11.

ACP is recruited to the ISC complex by ISD11 only and interacts via protein contacts (including the LYR motif) and fatty acyl chain insertion into the ISD11 hydrophobic tunnel. The latter interaction may depend on the chain length and thereby link the efficiency of Fe/S cluster biosynthesis to lipid and energy metabolism. In fact, our E. coli expression studies provide a first clue that the extent of ACP binding to NFS1-ISD11 may depend on the energetic situation of the cell. Growth conditions favoring high acetyl-CoA amounts and hence longer fatty acyl chains on ACP yielded significant NFS1-ISD11-ACP binding, while conditions with low acetyl-CoA levels and thus shorter fatty acyl chains favored negligible ACP association. In yeast, the different deletion phenotypes of ACP1 and mitochondrial fatty acid biosynthesis genes indicates a more complex situation which needs further investigation[25]. The dissociation of ISD11-ACP potentially caused by shorter fatty acyl chains or the action of Cys (Fig. 5e) may have consequences for the stability and/or functionality of NFS1. Extended absence of Isd11 or Acp1 in yeast mutants leads to aggregation of Nfs1 explaining their crucial in vivo role in Fe/S cluster synthesis. In vitro, however, both Isd11 and Acp1 were found dispensable for Nfs1 activity as long as Nfs1 is kept in solution (Fig. 4h, i)[47, 48]. This may well explain the observed differences for Isd11-Acp1 function in vivo and in vitro.

Since our structures contain bacterial ACP, the question arises, whether it is representative of mitochondrial ACP. This seems likely because ACPs are highly conserved during evolution. In particular, all ACP residues that contact ISD11 are conserved between E. coli and human explaining why bacterial ACP is co-isolated. Moreover, the mode of association of yeast mitochondrial Acp1 with Isd11 was confirmed by our mutational and biochemical studies. Therefore, the various ACP-containing ISC

complex structures determined here accurately describe the mitochondrial complexes. This view is corroborated by the structural similarity of ISD11-ACP and two LYRM-ACP sub-complexes present as supernumerary subunits in mitochondrial respiratory complexes I[22–24]. Based on our work, we propose that these sub-complexes may perform similar roles in complex I stabilization and/or regulation. The association of ACP with several crucial mitochondrial processes via LYRM proteins[21] underlines LYRM-ACP's role as a regulatory device that may coordinate the cellular energy status via mitochondrial fatty acid metabolism with Fe/S cluster generation, electron transport and ATP production. In summary, our 3D structures of the core ISC complex open numerous avenues of investigations on structural, mechanistic, and regulatory aspects of mitochondrial Fe/S cluster synthesis.

While this paper was submitted, a crystal structure of human NFS1-ISD11 with bound E. coli ACP was published[49]. This complex contains also two copies of each component, and the NIA sub-complex is similar to our structure. However, the two $(NIA)_2$ quaternary structures are strikingly different. In the $(NIA)_2$ structure presented by Cory et al. the NFS1 monomers make almost no contacts with their counterparts, a feature never observed in other known structures of PLP-dependent transaminases or desulfurases, which always form dimers like the one presented here. The biological and functional significance of the highly unusual structure described by Cory et al.[49] remains to be determined.

## Methods

**Protein expression and purification for crystallography.** The plasmid pZM2 harboring human *NFS1* encoding residues 56–457 ($\Delta$1-55) with a N-terminal His$_6$-tag and plasmid pZM4 containing full-length *ISD11* (no tag) were gifts from Silke Leimkühler (Molekulare Enzymologie, Universität Potsdam, Germany; Supplementary Table 3). The *NFS1* and *ISD11* genes were co-expressed in One Shot BL21 Star (DE3) cells (ThermoFisher Scientific) by selection on solid LB-Agar media containing ampicillin (100 µg ml$^{-1}$ and chloramphenicol (30 µg ml$^{-1}$). An overnight culture (6 ml, 37 °C) was used to inoculate 2 L of Terrific Broth (TB) medium (Fisher) containing the two selection antibiotics and supplemented with 10 µM PLP (Sigma-Aldrich) and 10 µM FeSO$_4$. After growth at 37 °C until OD$_{600}$ 0.6–0.8, the temperature was lowered to 22 °C. Protein expression was induced by 1 mM IPTG and cultured overnight. For expression of ISCU, the plasmid p24ac (gift from Dr. Kuanyu Li, Nanjing University, China; Supplementary Table 3) was used encoding ISCU1 residues 2–142 with a C-terminal His$_6$-tag[50]. The protein sequence was altered in position 107 to yield ISCU1-M107I protein (termed ISCU; plasmid p24ISCU_MI) that was described before as FXN-independent[32], in the expectation that frataxin would be dispensable as part of the core ISC complex. Cells were harvested by centrifugation for 20 min at 9000×g. The cell pellet was resuspended in purification buffer (10 mM BIS-TRIS pH 5.5, 200 mM NaCl, 20 mM KCl, 2 mM NaH$_2$PO$_4$, 2 mM Na$_2$HPO$_4$, 5% glycerol) and flash frozen in liquid nitrogen for storage at −80 °C.

The cell pellet containing co-expressed NFS1 and ISD11 was thawed at 42 °C and supplemented with 0.1 mM phenylmethylsulfonyl fluoride and 10 µM PLP. Resuspended cells were disrupted using the Constant Cell Disruption System (Constant System LTD., UK) under 35 MPa pressure. Cell lysate was spun down for 30 min at 20,000×g at 4 °C in Beckman Avanti J-26XP centrifuge. The supernatant was loaded on a Ni-NTA column (GE Healthcare) using Bio-Rad NGC FPLC system with an integrated sample pump. The column was washed with 20 column volumes of purification buffer supplemented with imidazole in a 0–75 mM gradient. The proteins were eluted with 2 column volumes of 125 mM imidazole in purification buffer containing 1 mM 1,4-dithiothreitol (DTT). Soluble NFS1-ISD11 complex was eluted as a single peak as followed by absorbance at 280 nm (aromatic amino acid residues) and 420 nm (PLP). Mass spectrometry of the purified protein fraction indicated, in addition to NFS1 and ISD11, the unexpected presence of E. coli ACP. The human NFS1-ISD11-ACP complex (termed $(NIA)_2$) was concentrated to 12–15 mg ml$^{-1}$, vitrified in liquid nitrogen, and stored in −80 °C.

To purify the human NFS1-ISD11-ACP-ISCU complex (termed $(NIAU)_2$), plasmid pZM2, encoding NFS1 with N-terminal His$_6$-tag, was modified to yield a *NFS1* construct (56–457) without tag (Supplementary Tables 3 and 4). Proteins NFS1-ISD11 (no tag) and ISCU-His$_6$ were expressed in separate cells, and pellets were combined prior to purification, which followed the standard protocol for isolation of the $(NIA)_2$ complex. The $(NIAU)_2$ complex was eluted from the Ni-NTA column with 75 mM imidazole in purification buffer, and was immediately

supplemented with 1 mM DTT and with or without 1 mM EDTA. Proteins were concentrated to 17–20 mg ml$^{-1}$ and vitrified in liquid nitrogen.

**Dynamic Light Scattering and Multi Angle Light Scattering**. Prior to crystallization experiments the polydispersity of the human (NIA)$_2$ complex in solution was measured using Wyatt DLS Plate Reader II at 826.7 nm at a range of temperatures. The sample and buffer were pre-filtered using 0.22 µm syringe filter and 50 µl of solution was injected into a measurement well in 384-well plate from Greiner Bio-One (Monroe, NC, USA). The average polydispersity (%Pd) of the purified (NIA)$_2$ complex was calculated using Dynamics V7 software. The best results were obtained at temperatures around 12 °C and the estimated polydispersity was in the 15–20% range. The average molecular masses of the complexes were determined using a multi-angle light scattering using size exclusion chromatography column (Wyatt SEC WTC 030S5) connected to GE Healthcare FPLC ÄKTA system in tandem with miniDAWN TREOS and Wyatt refractometer Optilab tREX. Fractions were collected and protein purity was analyzed by sodium dodecyl sulfate polyacrylamide gel electrophoresis (SDS-PAGE). Molecular mass of the human (NIA)$_2$ complex estimated as 125–130 kDa by Wyatt ASTRA 6 software, which suggested a dimeric state (calculated MM$_{th}$ = 146.7 kDa). The (NIAU)$_2$ complex was subjected to the crystallization screening directly after size exclusion chromatography, without evaluating polydispersity.

**Circular Dichroism**. The secondary structure and folding of the (NIA)$_2$ complex was investigated by CD using Chirascan Plus CD Spectrometer (Applied Photophysics, UK). Purified complex was diluted into buffer (10 mM BIS-TRIS pH 6, 50 mM NaCl), and the CD spectrum was measured between 280–195 nm at 20 °C. The spectrum was analyzed using Chirascan software to determine the secondary structure content. CD was also used to measure thermal stability of the complex. Spectra were measured at temperatures from 5–90 °C and analyzed with the Global3 software yielding a melting temperature of the complex between 25–30 °C.

**Mass spectrometry of gel fragments containing proteins**. Mass spectrometry analysis of the proteins separated on 15% SDS-PAGE gel was performed as follows. Gel fragments containing proteins were resuspended in 20 µL trypsin buffer (trypsin in 1 mM HCl and 200 mM NH$_4$HCO$_3$) followed by addition of 30 µL of 200 mM NH$_4$HCO$_3$ to each sample. Samples were incubated at 30 °C with shaking at 300 rpm. Trypsin digestion reaction was quenched by adding 1% trifluoroacetic acid (TFA) and tryptic peptides were extracted from gel samples using 100 µL of 0.1% TFA in 60% acetonitrile (ACN) and dried in the speed vac. Peptides were reconstituted in 10 µL of MS grade water:ACN:formic acid (97:3:0.1 v v$^{-1}$) and analyzed by liquid chromatography-tandem mass spectrometry (LC-MS/MS). Spectral results were collected over a mass range of 250–1700 (mass∗charge$^{-1}$; m z$^{-1}$) at a scan rate of 8 spectra per s. MS/MS data were collected over a range of 50–1700 m z$^{-1}$ and a set isolation width of 1.3 atomic mass units. Spectral data were converted to a mass∗charge$^{-1}$ data format using Agilent MassHunter Qualitative Analysis Software (Agilent Technologies Canada Ltd., Mississauga, ON, CA) and were processed against the UniProt *Escherichia coli* database, using SpectrumMill (Agilent Technologies Canada Ltd., Mississauga, ON, CA) as the database search engine. Search parameters included a fragment mass error of 50 ppm, a parent mass error of 20 ppm, trypsin cleavage specificity, and carbamidomethyl as a fixed modification of cysteine and oxidized methionine as a variable modification.

**Mass spectrometry of proteins in solution**. Protein solutions containing purified (NIA)2 or (NI)2 were digested by the addition of Sequencing Grade Modified Trypsin (Promega) and incubated at 37 °C overnight. The mass spectrometric analysis of the samples was performed using an Orbitrap Velos Pro mass spectrometer (ThermoScientific). An Ultimate nanoRSLC-HPLC system (Dionex), equipped with a custom 20 cm × 75 µm C18 RP column filled with 1.7 µm beads was connected online to the mass spectrometer through a Proxeon nanospray source. 1–15 µL of the tryptic digest (depending on sample concentration) were injected onto a C18 pre-concentration column. Automated trapping and desalting of the sample was performed at a flow rate of 6 µL min$^{-1}$ using water/0.05% formic acid as solvent. Separation of the tryptic peptides was achieved with the following gradient of water/0.05% formic acid (solvent A) and 80% ACN/0.045% formic acid (solvent B) at a flow rate of 300 nL min$^{-1}$: holding 4% B for 5 min, followed by a linear gradient to 45%B within 30 min and linear increase to 95% solvent B in additional 5 min. The column was connected to a stainless steel nanoemitter (Proxeon, Denmark)) and the eluent was sprayed directly towards the heated capillary of the mass spectrometer using a potential of 2300 V. A survey scan with a resolution of 60,000 within the Orbitrap mass analyzer was combined with at least three data-dependent MS/MS scans with dynamic exclusion for 30 s either using CID with the linear ion-trap or using HCD combined with orbitrap detection at a resolution of 7500. Data analysis was performed using Proteome Discoverer (ThermoScientific) with SEQUEST and MASCOT (version 2.2; Matrix science) search engines using either SwissProt or NCBI databases. For quantitation of the protein abundance the "integrated peak area" of the spectra was used.

**Protein crystallization**. The crystallization screens for the (NIA)$_2$ complex were dispensed into 96-well sitting drop plate using Gryphon crystallization robot

(ArtRobbins Instruments, Sunnyvale, CA). Each drop contained 300 nL of the complex concentrated to 12 mg ml$^{-1}$ and 300 nL of the mother liquor. The initial crystals appeared at 12 °C in a well containing the Protein Complex suite (Qiagen, Toronto) solution 100 mM HEPES pH 7.5, 0.2 M ammonium acetate and 25% isopropanol. Optimized crystals were obtained from the wells containing 100 mM 2-(N-morpholino)ethanesulfonic acid (MES) pH 6.5, 0.3 M ammonium acetate, 20 mM calcium acetate, 20 mM CaCl$_2$, and 19% isopropanol. The crystallization plate was transferred to 4 °C, and the crystals were harvested immediately and stored in liquid nitrogen using 17% MPD as a cryo-protectant. The best crystals diffracted to 2.75 Å resolution, belonged to the trigonal system, space group R32, with cell dimensions $a$ = 140.8 Å, $b$ = 140.8 Å, $c$ = 203.3 Å, $\alpha = \beta = 90°$, $\gamma = 120°$, and contained one molecule of each protein in the asymmetric unit.

The (NIAU)$_2$ complex solution was concentrated to 17–20 mg ml$^{-1}$ and dispensed into 96-well sitting drop plate as described above. The initial crystals appeared at 20 °C in wells containing 0.1 M MES pH 6.5, 10% PEG-20,000 or 0.1 M HEPES pH 7.5, 20% PEG-8000 in Compas Suite (Qiagen, Toronto). The best diffracting crystals were grown in the optimized condition of 0.1 M MES pH 7.0, 15% PEG-3350 at 15 °C to initialize nucleation. After 3 h plates were transferred to 12 °C, and incubated until crystals were fully-grown. Crystals were harvested at 4 °C and stored in liquid nitrogen using 20% glycerol as a cryo-protectant. Crystals with Zn ((NIAU-Zn)$_2$ complex) diffracted to 3.3 Å, space group P2$_1$2$_1$2$_1$ with $a$ = 97.5 Å, $b$ = 121.2 Å, $c$ = 151.5 Å, and without Zn ((NIAU)$_2$ complex) to 3.15 Å, space group P2$_1$2$_1$2$_1$ with cell dimensions $a$ = 98.4 Å, $b$ = 123.3 Å, $c$ = 151.7 Å.

**Data collection and structure refinement**. Diffraction data were collected using 08ID beamline at the Canadian Light Source, CMCF Sector, Saskatoon, SK, using a Rayonix MX300 CCD detector. Data were integrated and scaled using XDS package[51]. Initial phases were derived by molecular replacement method with Phaser MR[52] using *E. coli* IscS cysteine desulfurase (PDB code 3LVM)[16] as a model. The ISD11 model was built into density using ARP/wARP[53]. Additional electron density was present in the map calculated at this stage. Based on the mass spectrometric identification of ACP as part of the complex the MolRep program[54] was used to place the *E. coli* ACP model (PDB code 2FAE,[31] and it fitted nicely into the unaccounted density. The structure was refined using the PHENIX package[55], and manual rebuilding was carried with COOT[56]. A strong density in the 2mFo-DFc and difference maps extended from the residue Ser36 of ACP in all structures and was modeled as a lipid molecule with a chain containing 12 carbon atoms. Similarly, the PLP molecule was clearly visible in all three structures and was included in the model.

The final model of the (NIA)$_2$ complex contains residues 65–84, 97–274, 295–359, 404–431 of NFS1, 3–77 of ISD11, and 3–73 of ACP. Three segments of NFS1 are disordered in the structure, namely 85–96, 275–294, and 360–403, the latter containing the active-site cysteine (Cys loop) (Supplementary Fig. 1d). The final $R_{work}$ and $R_{free}$ are 0.213 and 0.255. The model of (NIAU-Zn)$_2$ contains NFS1 residues 54–452 (chains A), 54–454 (chain E), ISD11 5–85 (chain B) and 3–85 (chain F), ACP 4–74 (chain C), and 3–72 (chain G), ISCU 6–133 (chain D), and 10–133 (chain H). The NFS1 chain E has a partially disordered Cys loop for residues Cys381-Ser383. The final $R_{work}$ and $R_{free}$ are 0.196 and 0.255. The (NIAU)$_2$ complex showed more distortion of the NFS1 chain A (C terminus). Both NFS1 chains lack residues of the flexible Cys loop (Ala381-Leu385) due to the lack of stabilization by Zn ions bound to ISCU. The ISCU chain D is less well ordered and while the main chain trace is sufficiently well defined, many side chains are only partially ordered and could not be reliably modeled. This model contains NFS1 residues 55–380 and 386–455 (chains A), 54–379 and 385–456 (chain E), ISD11 5–85 (chain B) and 4–85 (chain F), ACP 4–75 (chain C) and 3–73 (chain G), ISCU 15–132 (chain D) and 10–135 (chain H). The final $R_{work}$ and $R_{free}$ are 0.188 and 0.242. The model geometries were verified by MolProbity[57]. Data collection and refinement statistics are provided in Table 1.

**Protein expression and purification for biochemistry or SAXS**. CtNfs1-CtIsd11 (-His$_{10}$)-ACP, CtIsu1-His$_6$, or His$_6$-CtYfh1 from *C. thermophilum*, and the NFS1-ISD11(-His$_{10}$) complex from *H. sapiens* were expressed in *E. coli* and purified by Ni-Sepharose affinity chromatography and gel filtration as described previously[28]. Variations in liquid media led to different amounts of bound ACP. Human or *C. thermophilum* NFS1-ISD11(-His$_{10}$) complexes purified from cells grown in LB medium did not bind any detectable amounts of ACP, while purification from cells grown in TB medium yielded substantial amounts of bound ACP. This is likely due to different lipid chain lengths. Recombinant human FDX2[27] and ISCU2 (Supplementary Table 3) was purified by anion and cation exchange chromatography, respectively, and subsequent gel filtration as described previously.

**Reconstitution of de novo Fe/S cluster synthesis on ISCU**. In vitro reconstitution assays[6, 28] were prepared in a Coy anaerobic chamber using freshly prepared stock solutions. Protein solutions were stored under anaerobic conditions for at least 6 h prior to experiments. The 300 µL standard reaction contained 2.5 µM *C. thermophilum* or human NFS1-ISD11±ACP, 75 µM ISCU2, 3 µM FXN, 3 µM FDX2, 0.3 µM FdxR in reconstitution buffer (35 mM Tris pH 8.0, 150 mM NaCl, 200 µM MgCl$_2$, 300 µM FeCl$_2$, 800 µM Na-ascorbate, and 500 µM NADPH). The reaction was transferred to a CD cuvette, sealed tightly and incubated at 30 °C in a

## Table 1 X-Ray diffraction data and refinement statistics

| | NFS1-ISD11-ACP (NIA)$_2$ | NFS1-ISD11-ACP-ISCU1 (NIAU)$_2$ | NFS1-ISD11-ACP-ISCU1 + Zn$^{2+}$ (NIAU-Zn)$_2$ |
|---|---|---|---|
| Wavelength | 0.97857 | 0.97857 | 0.97857 |
| Resolution range | 32.1-2.75 (2.85-2.75) | 46.8-3.15 (3.26-3.15) | 48.85-3.30 (3.52-3.30) |
| Space group | R 3 2 | P 2$_1$ 2$_1$ 2$_1$ | P 2$_1$ 2$_1$ 2$_1$ |
| Unit cell (Å) | a = 140.8 | a = 98.4 | a = 97.5 |
| | b = 140.8 | b = 123.3 | b = 121.2 |
| | c = 203.3 | c = 151.7 | c = 151.5 |
| Measured reflect | 222,416 (22,855) | 296,745 (47,168) | 372,306 (58,612) |
| Unique reflections | 20,346 (2020) | 32,513 (3133) | 27,226 (2646) |
| Multiplicity | 10.9 (11.3) | 9.1 (9.3) | 13.7 (13.7) |
| Completeness (%) | 99.59 (99.95) | 99.7 (98.09) | 99.7 (98.4) |
| Mean I/sigma(I) | 20.3 (1.93) | 15.4 (2.98) | 13.7 (2.73) |
| Wilson B-factor | 86.3 | 79.23 | 78.9 |
| R-merge | 0.069 (1.35) | 0.145 (0.95) | 0.235 (1.25) |
| CC1/2 | 0.998 (0.788) | 0.998 (0.905) | 0.997 (0.87) |
| Refinement (# refl) | 20,346 (2020) | 32,498 (3130) | 27,204 (2644) |
| R-work | 0.213 (0.287) | 0.188 (0.294) | 0.196 (0.292) |
| R-free | 0.255 (0.329) | 0.242 (0.328) | 0.255 (0.372) |
| # of non-H atoms | 3247 | 9886 | 9865 |
| Macromolecules | 3198 | 9788 | 9765 |
| Ligands | 49 | 98 | 100 |
| Protein residues | 439 | 1348 | 1361 |
| RMS (bonds) | 0.004 | 0.003 | 0.004 |
| RMS (angles) | 0.66 | 0.56 | 0.58 |
| Ramachandran plot | | | |
| Favored (%) | 96 | 93.88 | 94.78 |
| Allowed (%) | 4.0 | 5.44 | 3.88 |
| Outliers (%) | 0 | 0.68 | 1.34 |
| Rotamer outliers (%) | 0.33 | 2.77 | 1.85 |
| Clashscore | 9.4 | 15.3 | 16.34 |
| Average B-factor | 99.96 | 89.74 | 83.38 |
| Macromolecules | 99.97 | 89.78 | 83.37 |
| Ligands | 99.22 | 85.95 | 84.38 |
| No. of TLS groups | 3 | 8 | 8 |
| PDB code | 5WGB | 5WKP | 5WLW |

CD spectrometer (Jasco, J-815). The CD-signal at 431 nm was recorded and after 2 min the reconstitution reaction was initiated by the addition of 500 µM cysteine. After 20 min a full spectrum was recorded from 300 nm to 650 nm.

**Protein interactions by Microscale Thermophoresis**. MST[58] was performed on a Monolith NT.115 (Nano Temper Technologies GmbH, Munich, Germany) at 21 °C (red LED power was set to 50% and infrared laser power to 75%). 20 µM human (NIA)$_2$ complex was labeled with the dye NT 647 supplied by Nano Temper Technologies. Labeled (NIA)$_2$ complex was titrated as indicated with ISCU2 in buffer T (35 mM KP$_i$ pH 7.4 and 150 mM NaCl). Prior to MST measurements ISCU2 was treated with 5 mM DTT, 1 mM KCN, and 10 mM EDTA and subsequently gel filtered using a HiLoad 16/60 Superdex 75 PG (GE Healthcare) either aerobically or anaerobically in a Coy anaerobic chamber. At least nine independent MST experiments were performed at 680 nm and processed by Nano Temper Analysis package 1.2.009 and Origin8 (OriginLab, Northampton, MA).

**Dissociation of the NFS1-ISD11-ACP complex**. 25 µM CtNfs1-CtIsd11-His$_{10}$-ACP in buffer D (35 mM Tris-HCl pH 7.4, 150 mM NaCl and 5% (v v$^{-1}$) glycerol) was anaerobically incubated for 4 h at 15 °C with either 15 mM cysteine or 15 mM serine or 3 mM DTT. Samples were bound to Ni-NTA agarose (IBA Life Science) and incubated aerobically at 4 °C for 1 h. Samples were centrifuged at 1500×g at 4 °C for 5 min, and the supernatants were analyzed by SDS-PAGE and Coomassie staining. The remaining Ni-NTA beads were incubated for 10 min with buffer D containing 250 mM imidazole, and again centrifuged as described above. The supernatants were analyzed as above.

**SAXS methodology and modeling of components in complexes**. All complexes and monomeric proteins were prepared or mixed in buffer S1 (35 mM Tris-HCl pH 7.4, 150 mM NaCl with or without 5 mM Cys) in a Coy anaerobic chamber. Protein solutions were subsequently subjected to size exclusion chromatography. Prior to final measurements monomeric ISC proteins and the various ISC complexes were desalted twice by passing them through a PD10 column (GE Healthcare) in buffer S1. Samples were subsequently cleared by ultracentrifugation (20,000×g, 20 min) and placed in the sample holder. SAXS data were collected at minimum three protein concentrations ranging from 1 to 10 mg ml$^{-1}$. The SAXS experiments were carried out on the BM29 BioSAXS beamline at ESRF (European Synchrotron Radiation Facility) equipped with a Pilatus 1 M detector at standard set-up. Ten frames were recorded for each concentration and sample.

SAXS data evaluation was carried out using the ATSAS software package (ATSAS 2.6.0)[59]. GNOM[60] was used to obtain the Pair distribution function ($P_r$) and $D_{max}$ values. The radius of gyration ($R_g$) was determined using the Guinier approximation. For calculation of 20 ab initio models the output of GNOM was used together with DAMMIF[61] and the respective 3D structures were subsequently fitted to the average shape using UCSF Chimera 1.11[62]. The location and orientation of CtFdx2 was adjusted to the known interaction surface within the ferredoxin-Isu1 heterodimer[28]. The location and orientation of CtYfh1 relative to CtNfs1 was based on NMR and SAXS solution structures of the bacterial counterparts (IscS-CyaY[36]) and the location of bacterial CyaY on IscS[16]. For final rigid body modeling of all proteins/complexes, SASREF was used[63]. Therefore, scattering amplitudes for human ISC complexes were calculated based on our crystal structures using CRYSOL[64] and compared to experimental data. For the *C. thermophilum* ISC proteins and complexes, we used homology models based on *E. coli* or yeast structures. All samples yielded good or excellent $\chi^2$ values.

**Analysis of ISD11 mutations in yeast and enzyme activities**. Wild-type and mutant versions of *S. cerevisiae* ISD11 (nucleotides 1–375, synthesized by GenScript, Piscataway, USA) with a C-terminal FLAG-tag were inserted into vector p416MET25 (Supplementary Table 3). *S. cerevisiae* GalL-ISD11 cells (W303-1A; *pISD11::pGALL-natNT2*; this work) harboring the different versions of p416-ISD11-FLAG and pRS414-ACP1-HA[25] (Supplementary Table 3) were cultivated in SD minimal medium for 40 h and mitochondria were isolated[65]. Mitochondria were lysed at a protein concentration of 1.5 µg µl$^{-1}$ for 5 min at 4 °C in buffer E (50 mM Tris-HCl (pH 8), 50 mM NaCl, 0.2% (v v$^{-1}$) Triton-X100). Debris was removed by centrifugation (20,000 ×g, 20 min at 4 °C). Enzyme activities were measured according to Pierik et al.[66]. Desulfurase activity determination of NFS1 was carried out by detecting sulfide production with methylene blue[67] using 20 µg of *C. thermophilum* or human purified NFS1-ISD11 with or without bound ACP. For determination of desulfurase activity in isolated mitochondria, 200 µg of mitochondrial lysates (see above) were used.

**Co-immunoprecipitation of mitochondrial proteins**. Mitochondria (250 µg of protein) were lysed for 5 min on ice in 250 µl buffer (10 mM Tris-HCl, pH 7.4, 150 mM NaCl, 0.1 mM DTT, 10 % v v$^{-1}$ glycerol, 0.2% v v$^{-1}$ Triton-X100) supplemented with 2 mM phenylmethylsulphonylfluoride. Membrane debris was removed by centrifugation (12000×g, 15 min, 4 °C) and the mitochondrial lysate was incubated with 30 µl FLAG-M2 Sepharose beads (Sigma Aldrich) for 1 h on a rotary shaker at 4 °C. Beads were pelleted by centrifugation (1000×g, 1 min, 4 °C) and washed 3 times with 500 µL buffer. Bound proteins were subjected to SDS polyacrylamide gel electrophoresis and identified by immunostaining using specific antibodies (α-Nfs1, α-Isu1, and α-porin: self-raised (1:5000; rabbit), α-HA: Santa Cruz (sc-7392) (1:15,000; mouse); α-FLAG: SIGMA (011M4789) (1:20,000; rabbit); secondary α-mouse: Bio-Rad (L1706516) (1:1000); secondary α-rabbit: Sigma-Aldrich (A1949) (1:1000)).

**Miscellaneous methods**. The following published methods were used: yeast cell technologies[68]; manipulation of DNA and PCR[69]; and immunological techniques[70]. Plasmids used in this study are listed in Supplementary Table 3. Primers used in this study are listed in Supplementary Table 4.

**Data availability**. Atomic coordinates and structure factors for ((NIA)$_2$)$_2$, ((NIAU)$_2$)$_2$, and (NIAU-Zn)$_2$ complexes were deposited at the Protein Data Bank (http://www.rcsb.org/pdb/home/home.do) with the codes 5WGB, 5WKP, and 5WLW, respectively. The data that support the findings of this study are available from the corresponding authors upon reasonable request.

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

## Acknowledgements

We thank Dr. G. Katselis for performing mass spectrometry analysis. M.C. acknowledges the support of the Canadian Institutes of Health Research (CIHR) (Grant MOP-48370) and from and Canadian Foundation for Innovation. R.L. acknowledges generous financial support from Deutsche Forschungsgemeinschaft (SPP 1927), the LOEWE program of state Hessen, and the Core Facilities of Protein Spectroscopy and Protein Biochemistry, and Mass Spectrometry of Philipps-Universität Marburg. We thank the staff of the beamline 08ID at the Canadian Light Source, which is supported by the NSERC, the National Research Council Canada (NRCC), the CIHR, the Province of Saskatchewan, Western Economic Diversification Canada, and the University of Saskatchewan and staff of the European Synchrotron Radiation Facility (ESRF), Grenoble, France. Molecular graphics and analyses were performed with the UCSF Chimera package. We gratefully acknowledge the use of instruments at the Protein Characterization and Crystallization Facility (PCCF), College of Medicine, University of Saskatchewan.

## Author contributions

M.T.B., S.A.F., R.L., and M.C. planned the experiments and analyzed the data. The crystallographic work was performed by M.T.B.; the SAXS and biochemical analyses and the art work was done by S.A.F.; yeast work was assisted by U.M. The manuscript was written by R.L. and M.C., and revised by all authors.

## Additional information

**Competing interests:** The authors declare no competing financial interests.

