## [Peer Review file · Nature Communications]

Reviewers' comments:

Reviewer #1 (Remarks to the Author):

Boniecki et al. report on crystal structures of a (NSF1-ISD11-ACP)₂ complex and two forms of a (NSF1-ISD11-ACP-ISCU)₂ complex, one of them with Zn²⁺ bound. In addition, the authors report SAXS envelopes for several complexes including the especially relevant (NFS1-ISCU-FDX2-FXN)₂ species and fit the corresponding 3D structures within them. Overall, the results are interesting and add significant new data concerning Fe-S cluster biosynthesis. There are however some problems that need to be dealt with before the manuscript is fit for publication.

Abstract: "eukaryote-specific ACP"? The ACP in this study came from *E. coli*. "Three different NSF1-ISD11-ACP-ISCU complexes". This is incorrect. One of the complexes does not have ISCU.

Introduction: It should be mentioned here that the structure in ref. 16 has a bound [2Fe-2S] cluster at the expected site. Expressions like "putative" assembly site and "potential" ligands in pages 5 and 6 do not take into account the results reported in that reference.

Results: Crystal structures: The differences between *R*_{work} and *R*_{free} values in Supplementary Table 1 show that the structures are over-refined. One may also wonder what is the use of reporting a Wilson B-factor at these rather low resolutions. A section of the structures with matching electron density should be reported in the main text.

The authors make no reference to previously studied ISCU-Zn complexes (such as Liu et al. *Proteins*, 59, 875, 2005 and several papers by Pastore et al.). A comparison of their ISCU-Zn structure with previously reported ones has to be included. The same applies to their observation of a Zn-Cys381 bond. In ref. 16 Marinoni et al. found that the equivalent IscS active site Cys was a ligand to the stabilized [2Fe-2S] cluster in the *A. fulgidus* complex. Although the ISCU-Zn complex is probably an artifact this coincidence has to be discussed. Also a stereo figure of the superposition of the two structures should be included in the main text (replacing Supplementary fig. 1c which is almost impossible to figure out; the Zn and [2Fe-2S] cluster should be included and clearly visible).

ACP: Why didn't the authors perform a mass spectrometric analysis of their ACP-FA after dissociation from the complex? In this way they could have determine the exact nature of the bound fatty acid and figure out how this molecule is metabolized in mitochondria.

Mutagenesis (page 8): A significant part of this section could be moved to the Supplementary Information as many of the reported results are a confirmation of previous observations.

"Solution structures" (page 10): In fact, strictly speaking these are not solution structures but envelopes where you can fit crystal or NMR structures.

"bacterial IscS-IscU complex¹⁶" (page 13): *A. fulgidus* is an archeon not a bacterium.

"...ISCU binding dynamically changes the NFS1 structure..." (page 14): the authors should add: "like in the case of the IscS-IscU complex from *A. fulgidus*".

"This behavior is strikingly different from the bacterial complex..."(page 13): There is a problem here. On the one hand the authors cite ref. 39 where CyaY and Fdx are proposed to bind to overlapping sites and, on the other hand, they use IscS-CyaY and Yah1-Isu1 structural data to show that frataxin and ferredoxin bind to different sites in their complex. This contradiction needs

to be discussed.

Reviewer #2 (Remarks to the Author):

The paper NCOMMS-17-14592-T entitled « Structure, function and dynamics of the mitochondrial Fe/S cluster synthesis complex » by Boniecki and collaborators describes the crystal and solution structures of various complexes involved in the mitochondrial Fe/S cluster biosynthetic pathway. Coupled to some in vitro biochemical and yeast functional analyses, the authors provide mechanistic and structural insights into the first steps of de novo Fe/S cluster synthesis. Using Human or *Chaetomium thermophilum* proteins, the authors showed evidence that NFS1 forms a tight complex with ISCU, FDX2, and FXN defining the catalytic center for the de novo Fe/S cluster synthesis. As previously observed, ISCU binding to NFS1 generates substantial structural rearrangements in NFS1 by rigidifying some disordered regions including the loop containing the catalytic cysteine. The results also point to the fact that both FDX2 and FXN can bind simultaneously to this complex in marked contrast with the bacterial model. Additional proteins, ISD11 and the associated ACP protein recently discovered, belong to this complex (at least transiently) but they would not have catalytic roles, rather acting in NFS1 stabilization and possibly other yet unknown regulatory functions. All subunits are present in a stoichiometric amount around a central Nfs1 dimer. Each ISD11 monomer contacts both NFS1 monomers using mostly polar-ionic interactions and hydrogen bonds but also hydrophobic interactions. ACP binds solely via ISD11 and the binding seems dependent on the chain length of the fatty acyl chain which comes into a hydrophobic tunnel found on ISD11. The observation that adding cysteine to a ternary NFS1-ISD11-ACP complex led to the dissociation of ISD11-ACP from the NFS1 dimer is particularly interesting, thinking to the dynamic nature of this system. Although its depletion leads to Fe/S protein biogenesis defects (van Vranken et al 2016), ACP is dispensable for in vitro Fe-S cluster reconstitution assays raising questions about its role. To sum up, the study is well designed and brings substantially new molecular and structural details about the eukaryotic Fe-S cluster assembly machinery even though a very similar study came out a few days ago (Cory SA, Van Vranken JG, Brignole EJ, Patra S, Winge DR, Drennan CL, Rutter J, Barondeau DP. Structure of human Fe-S assembly subcomplex reveals unexpected cysteine desulfurase architecture and acyl-ACP-ISD11 interactions. *Proc Natl Acad Sci U S A*. 2017 Jul 3;114(27): E5325-E5334. doi: 10.1073/pnas.1702849114). Then, my major comment will be that the authors should consider introducing some elements of comparison with the data presented in the latter paper.

I listed below a few other concerns that need to be addressed.

1. There could be somewhere in the supplementary information a figure of NFS1 structure with labelled secondary structures, so that it is easier to follow the description of the position of the flexible loops but also of the structural rearrangements occurring on NFS1.
2. In fig 1a/b/c, it seems that the number of secondary structures in ACP varies. Is it simply a difference of resolution or could this discrepancy be solved by additional rounds of refinement ?
3. In Fig 2b, the arrow corresponding to Cys44 ISCU may be better positioned. Since it is discussed in the text, the position of Trp454 of NFS1 could be shown here.
4. In the SAXS experiments performed with *Chaetomium* enzymes, it is surprising that Isd11 is not represented in any of the complexes shown in fig. 6 whereas it is indicated in the methods that it was co-expressed with other proteins. I understood from the figure S5 that there might be some repurification steps to remove ISD11 and ACP1. Incidentally this indicates that NFS1 is stable without ISD11. All of this may be mentioned somewhere in the text.
5. p19 : Define the c in cISCU1. I am surprised that the full-length sequence is used for heterologous expression. Is there no cleaved targeting sequence on this protein ?
6. p23 : in the sentence « Two segments of NFS1 are disordered in the structure, ... », one should probably read three instead of two.
7. p29. In the legend of fig 4, maybe indicate that Acp1-HA protein expressed is from yeast since

most of the structural rely on the *E. coli* counterpart.

8. In the legend of figure 5, it is indicated that ISCU was used aerobically or anaerobically. This formulation is not adequate. My understanding is that the experiment was eventually achieved under anaerobic condition but which kind of protein has been used ? Is it an apo or holo ISCU ? Since I have the feeling that this is an apo, does it have zinc or not ? has it been reduced and desalted before MST experiments to keep the protein reduced ? Adding these details is important for sustaining the conclusions made in the text in p11.

Reviewer #3 (Remarks to the Author):

The structural models reported here represent highly significant advances in our understanding of the biosynthesis of iron-sulfur proteins in mitochondria. The structural difference in the NFS1 dimer interface before and after the addition of ISCU, is very interesting, in particular because of the recent publication of a structural model for NIA based on X-ray crystallography with a very much different NFS1 dimer interface.¹ Although the complex containing Zn²⁺-bound ISCU is non-physiological, its structure is instructive in that the Cys-loop of NFS1 is resolved and shown to extend toward ISCU. The evidence presented for a conformational effect of added L-cysteine is also of importance, and should inspire follow-on experiments.

This reviewer thanks the authors for providing preliminary PDB validation reports for the three structures. It is regrettable, however, that the structures had not been deposited in the PDB so that these reports could have been vetted by PDB annotation.

The manuscript suffers from several issues that need to be corrected

1. When introducing organisms, the full names need to be provided: *Chaetomium thermophilum*, etc.
2. The statement on p. 3 that "the ISC machinery ... [in mitochondria]...consists of 18 proteins of bacterial origin" seems bit misleading and might benefit from rephrasing.
3. The same molecular abbreviations must not be used for homologous proteins or protein variants: in the manuscript, ISCU is used interchangeably for ISCU (wild type), ISCU(M108I), and Isu1. This is highly confusing. The manuscript should use explicit names for variants along with the names of the *C. thermophilum* proteins used in the authors' previous publication to avoid confusion.
4. The authors report that complexes lacking Acp have activity identical to complexes with added Acp. Have the authors rigorously determined the absence of Acp in complexes "lacking Acp". This needs to be done by MS analysis. Activity in the absence of Acp would seem to contradict the hypothesis that Acp is required and that acyl-Acp regulates activity.
5. A "regulatory" role for ISD11 is stressed over and over in the manuscript without much justification, aside from its not being "catalytic". Could it not also play a structural role, as indicated by studies showing that NFS1 is less stable in its absence?
6. The rationale for using ISCU(M108I) rather than wild-type ISCU should be mentioned.
7. In the ONLINE METHODS, it is unclear at times when NFS1 is being produced with a His-tag and when it is not.

[1] Cory, S. A., Van Vranken, J. G., Brignole, E. J., Patra, S., Winge, D. R., Drennan, C. L., Rutter, J., and Barondeau, D. P. (2017) Structure of human Fe-S assembly subcomplex reveals unexpected cysteine desulfurase architecture and acyl-ACP-ISD11 interactions, *Proc. Natl. Acad. Sci. U. S. A.* 114, E5325-E5334.

Reviewer #4 (Remarks to the Author):

Summary:

This article focus on the structure and function of NFS1-ISCU. This complex is involved in the in de novo synthesis of a Fe/S cluster, and more particularly, in the supply of sulfur. The structures of the complex and different sub-complexes have been resolved and are presented. The structural role of several residues at the interface between the different components or in the vicinity of binding site is discussed. The role of these residues is explored in vivo and confirmed by monitoring the growth, isd11 interaction, enzyme activity of different mutant. Finally the complexes are studied in solution using small angle scattering.

My field of expertise being small angle X-ray scattering, I cannot properly assess the impact and originality of the work and my comments will focus on the SAXS part.

The SAXS data and their interpretation are convincing and support the claims of the author, however, a couple of points could be improved or specified:

- "Similar results were obtained by both species (Fig. 5a-b; supplementary Fig 5a-b and Table 2 and 3)". Plots and model of different species are on different figures and are therefore difficult to compare. The Rg of the different species reported in the table also differs between the species. It should be better explained why the results from the two species are similar.
- In sup table 3 the Rg is given with significant digit up to the picometer. Rounding the Rg to the tenth of Angstrom like in sup table 2 (or even angstrom) would be more appropriate. The error should also be accordingly modified (together with the errors on the radius of gyration of (NFS1-FDX2)₂, (NFS1-FDX2-FXN)₂, (NFS1-ISCU-FDX2-FXN)₂ from sup table 2).
- For both species, the (NIA)₂ data were fitted and compared to crystallographic structure of (NIA)₂ alone and (NIA)₂ sub-complex taken from the (NIAU)₂ crystal. The agreement of the crystallographic structure with the SAXS data is illustrated using the ab initio shape and the chi² of the fit is given in the text. While ab initio model can provide the overall shape and are good for illustration, the fits of the SAXS curves computed from the crystallographic structure directly to the experimental data are much more convincing and could be shown in a figure.
- Along the same line, for the human (NIA)₂ (figure 5d), it would be much more convincing to underline directly the difference between the curves computed from the crystallographic structure and the experimental data rather than between the ab initio model and the crystal structure (with the red circle). (Ab initio models are build assuming a homogeneous electron density protein, which results in a loss of resolution.)
- It is not clear which ab initio shape are shown in the figures? From the methods section, one can understand that 20 models have been built but which one is represented, the most typical, the average, a refined shape?
- In the part solution structures of NFS1, ISCU, FDX2 and FXN sub-complexes, it is not clear from the text if all the models of the complexes are obtained by rigid body modelling (with sasref). (+ If this is the case, and if the positions of the different monomer are the same in the different complexes, a rigid body approach where the SAXS curves of the different complex would be simultaneously fitted and the models of the different complex simultaneously built (imposing known interfaces between monomers if needed) could lead to more robust models.)

Response to Reviewers' comments

Manuscript No. NCOMMS-17-14592-T (Boniecki et al.)

We first would like to thank the four Reviewers for their careful reading of our manuscript. We are glad to have received four overall positive evaluations. The constructive suggestions of the Reviewers have helped us to further improve our manuscript, and we hope it is now suitable for publication. Our point-to-point response is presented in detail below. We also used the revision to further polish the text of our manuscript.

Note: *Reviewer's text is in italics (shortened to the questions and concerns); our response is in plain text. All changes during revision are highlighted in another submitted file.*

Reviewer #1

.....

Abstract: "eukaryote-specific ACP"? The ACP in this study came from E. coli. "Three different NFS1-ISD11-ACP-ISCU complexes". This is incorrect. One of the complexes does not have ISCU.

At the position where "eukaryote-specific ACP" was used, the statement was correct. We have nevertheless removed it because *E. coli* ACP also was shown to bind IscS. We have changed the text in the Abstract to "three different NFS1-ISD11-ACP complexes with and without ISCU".

Introduction: It should be mentioned here that the structure in ref. 16 has a bound [2Fe-2S] cluster at the expected site. Expressions like "putative" assembly site and "potential" ligands in pages 5 and 6 do not take into account the results reported in that reference.

We have added a sentence to the Introduction to mention the presence of a [2Fe-2S] in one of the two IscS-IscU structures. "Putative" was used on purpose to reflect the fact that the IscS-like protein in this complex is not a bona fide cysteine desulfurase, as reported by Dr. Fontecilla-Camps in a later publication (BBA 1853, p. 1457, 2015). Nevertheless, we have changed it to "expected". The sentence containing "potential" has been rewritten to meet the reviewers critique.

Results: Crystal structures: The differences between Rwork and Rfree values in Supplementary Table 1 show that the structures are over-refined. One may also wonder what is the use of reporting a Wilson B-factor at these rather low resolutions. A section of the structures with matching electron density should be reported in the main text.

The structures were refined with phenix.refine with individual B-factors using "optimize X-ray/stereochemistry weight" and "optimize X-ray/ADP weight". We have tested in parallel refining with group B-factors, one or two per residue. The difference between the R-free and R-work decreased only marginally when group B-factors were used, from ~5.5% to ~5%, with a concomitant increase of R-free by 0.7-1.0%. Using group B-factors led to appearance of negative density in the difference maps at the ends of long sidechains, while when using individual B-factors no such negative density was observed. We therefore deposited the structures refined with individual B-factors. A detail of the ISD11 structure including the fatty acyl chain with matching electron density is now shown in Fig. S3d.

The authors make no reference to previously studied ISCU-Zn complexes (such as Liu et al. Proteins, 59, 875, 2005 and several papers by Pastore et al.). A comparison of their ISCU-Zn structure with previously reported ones has to be included. The same applies to their observation of a Zn-Cys381 bond. In ref. 16 Marinoni et al. found that the equivalent IscS active site Cys was a ligand to the stabilized [2Fe-2S] cluster in the A. fulgidus complex. Although the ISCU-Zn complex is probably an artifact this coincidence has to be discussed. Also a stereo figure of the superposition of the two structures should be included in the main text (replacing Supplementary fig. 1c which is almost impossible to figure out; the Zn and [2Fe-2S] cluster should be included and clearly visible).

These ISCU-Zn publications were indeed not cited because Nature Communications has a limit of 70 references. As suggested, a stereo view of the superimposition of the bacterial and human IscS-IscU parts (with Fe/S or Zn) is now presented as suggested (new Fig. S2a). We now show also a comparison of the Zn-bound bacterial and human IscU proteins in the Supplement (new Fig. S2b) and cite the work by Liu et al.

We also have mentioned cases of Zn and Fe/S cluster binding to the same sites on proteins to better make clear that the Zn binding site may reflect the location of Fe/S cluster synthesis (Discussion, 2nd paragraph).

ACP: Why didn't the authors perform a mass spectrometric analysis of their ACP-FA after dissociation from the complex? In this way they could have determine the exact nature of the bound fatty acid and figure out how this molecule is metabolized in mitochondria.

Before submission, we had analyzed our protein complexes +/- ACP by both mass spectrometry and SDS-PAGE, but did not feel the necessity to include these data sets. We now show the mass spectrometry data for protein complexes +/- ACP (new Fig. 4i). Additionally, we have analyzed by mass spectrometry dissolved crystals. The experimentally determined molecular mass of ACP with fatty acyl chain correspond to the molecular mass calculated for the 14 carbon chain, however there is interpretable density for only 12 carbons. This data is now included in the manuscript (chapter "Structure of ACP and its interaction with ISD11").

Mutagenesis (page 8): A significant part of this section could be moved to the Supplementary Information as many of the reported results are a confirmation of previous observations.

We respectfully disagree here. All this data on ISD11 mutagenesis (Fig. 4 and Fig. S4) and its comprehensive biochemical analysis by several methods is new.

"Solution structures" (page 10): In fact, strictly speaking these are not solution structures but envelopes where you can fit crystal or NMR structures.

We fully agree, even though the shapes (envelopes) represent (low resolution) solution structures. Nevertheless, we have changed the text to better reflect this point.

"bacterial IscS-IscU complex16" (page 13): A. fulgidus is an archeon not a bacterium.

Corrected.

...ISCU binding dynamically changes the NFS1 structure..." (page 14): the authors should add: "like in the case of the IscS-IscU complex from A. fulgidus".

The overlaid structures of the *A. fulgidus* IscS-like protein dimer (PDB code 4HVK) and this protein dimer in the structure of the IscS-IscU complex (PDB code 4EB5) superimpose with root-mean-squares-deviation of 1.19 Å and maintain the same dimer interface. This suggests

that the conformation of these IscS-like dimers is very similar (see Figure below; for reviewer only). We are not aware of other data showing changes in dimer conformation of *A. fulgidus* IscS-like protein upon association with IscU.

“This behavior is strikingly different from the bacterial complex...”(page 13): There is a problem here. On the one hand the authors cite ref. 39 where CyaY and Fdx are proposed to bind to overlapping sites and, on the other hand, they use IscS-CyaY and Yah1-IscU structural data to show that frataxin and ferredoxin bind to different sites in their complex. This contradiction needs to be discussed.

It is correct that we did not discuss the discrepancy. The reason is that our new structural data are fully consistent with our reconstitution assays (Webert et al. 2014; Freibert et al. 2017) in which all mitochondrial core ISC proteins are necessary for Fe/S cluster synthesis on ISCU. In bacteria, however, there is the discrepancy that CyaY (i.e. bacterial frataxin) inhibits the in vitro Fe/S cluster synthesis (Pastore), whereas it is positively needed in vivo (Barras' recent work). We still think that these complicated issues are better discussed in detail in a Review, and not in our original article on the mitochondrial ISC system (also because of size limitation here).

Reviewer #2

.....

To sum up, the study is well designed and brings substantially new molecular and structural details about the eukaryotic Fe-S cluster assembly machinery eventhough a very similar study came out a few days ago (Cory SA, Van Vranken JG, Brignole EJ, Patra S, Winge DR, Drennan CL, Rutter J, Barondeau DP. Structure of human Fe-S assembly subcomplex reveals unexpected cysteine desulfurase architecture and acyl-ACP-ISD11 interactions. Proc Natl Acad Sci U S A. 2017 Jul 3;114(27): E5325-E5334. doi: 10.1073/pnas.1702849114). Then, my major comment will be that the authors should consider introducing some elements of comparison with the data presented in the latter paper.

This paper by the Barondeau group came out after submission of our manuscript. It shows quite an unusual quaternary structure in which the typical NFS1 dimer is disrupted and turned by ca. 180° (possibly due to crystallization in the presence of acetone??). We now have discussed this paper (last paragraph of Discussion) and point out that all other known structures of this large class of PLP-dependent proteins (including transaminases) are similar to our dimeric structure. Proof for the physiological relevance of the unusual NFS1 structure remains to be provided.

I listed below a few other concerns that need to be addressed.

1. There could be somewhere in the supplementary information a figure of NFS1 structure with labelled secondary structures, so that it is easier to follow the description of the position of the flexible loops but also of the structural rearrangements occurring on NFS1.

A good suggestion. We now provide this information as new Fig. S1c and d.

2. In fig 1a/b/c, it seems that the number of secondary structures in ACP varies. Is it simply a difference of resolution or could this discrepancy be solved by additional rounds of refinement ?

We have redrawn Fig. 1 a-c with the coordinates as deposited in PDB. The ACP structures are virtually identical in all structures (Supplementary Fig. 1a).

3. In Fig 2b, the arrow corresponding to Cys44 ISCU may be better positioned. Since it is discussed in the text, the position of Trp454 of NFS1 could be shown here.

In the improved Fig. 2b, the arrow indicating Cys44 has been moved to clearly indicate this residue. We show here the NFS1 portion from the Zn complex because thereby one can better see ISCU's Ala-loop. However, Trp454 is not resolved in our structures (see Fig. S1d).

4. In the SAXS experiments performed with Chaetomium enzymes, it is surprising that Isd11 is not represented in any of the complexes shown in fig. 6 whereas it is indicated in the methods that it was co-expressed with other proteins. I understood from the figure S5 that there might be some repurification steps to remove ISD11 and ACP1. Incidentally this indicates that NFS1 is stable without ISD11. All of this may be mentioned somewhere in the text.

The procedure has been explained in Methods (section: SAXS methodology and ...). We further improved the text to make the sample treatment clearer. The loss of ISD11-ACP is caused by its dissociation from NFS1 in the presence of cysteine (as outlined in Fig. 5e). The fact that, in principle, re-isolated NFS1 is stable for some time without ISD11 is undisputed, because NFS1 desulfurase measurements were published by several groups

including our own before the discovery of ISD11. We have already cited these observations on p. 14, middle (“Previous results and our new data show that both ISD11 and ACP are not essential for cysteine desulfuration by NFS1, yet ...”).

5. p19 : *Define the c in cISCU1. I am surprised that the full-length sequence is used for heterologous expression. Is there no cleaved targeting sequence on this protein ?*

We have improved this information. We used the cytosolic version of ISCU (=ISCU1) which is virtually identical to mitochondrial ISCU2 minus the presequence.

6. p23 : *in the sentence « Two segments of NFS1 are disordered in the structure, ... », one should probably read three instead of two.*

Indeed, there are three disordered segments as shown by the numbers in the text. The mistake was corrected.

7. p29. *In the legend of fig 4, maybe indicate that Acp1-HA protein expressed is from yeast since most of the structural rely on the E. coli counterpart.*

Done. The nomenclature “Acp1” was supposed to indicate the yeast protein, but we now added the term “S. cerevisiae” to make it clear and to avoid two times “yeast”.

8. *In the legend of figure 5, it is indicated that ISCU was used aerobically or anaerobically. This formulation is not adequate. My understanding is that the experiment was eventually achieved under anaerobic condition but which kind of protein has been used ? Is it an apo or holo ISCU ? Since I have the feeling that this is an apo, does it have zinc or not ? has it been reduced and desalted before MST experiments to keep the protein reduced ? Adding these details is important for sustaining the conclusions made in the text in p11.*

The statement “aerobically or anaerobically” only applies to Fig. 5f (i.e. the equilibrium titration by MST). We have improved the text in Methods to better describe the experimental design. For SAXS experiments ISCU was always treated with DTT, KCN and EDTA, but we have no direct indication if Zn was adventitiously bound or not.

Reviewer #3

.....

This reviewer thanks the authors for providing preliminary PDB validation reports for the three structures. It is regrettable, however, that the structures had not been deposited in the PDB so that these reports could have been vetted by PDB annotation.

All three structures have now been deposited in the PDB data base.

The manuscript suffers from several issues that need to be corrected

1. *When introducing organisms, the full names need to be provided: Chaetomium thermophilum, etc.*

Corrected.

2. *The statement on p. 3 that “the ISC machinery ... [in mitochondria]...consists of 18 proteins of bacterial origin” seems bit misleading and might benefit from rephrasing.*

We rephrased the beginning of the Introduction to better indicate the evolutionary conservation and bacterial origin of the mitochondrial ISC machinery.

3. *The same molecular abbreviations must not be used for homologous proteins or protein variants: in the manuscript, ISCU is used interchangeably for ISCU (wild type), ISCU(M108I), and Isu1. This is highly confusing. The manuscript should use explicit names for variants along with the names of the C. thermophilum proteins used in the authors' previous publication to avoid confusion.*

We have defined where the human ISCU mutant protein was used, i.e. only for crystallography and SAXS. The *C. thermophilum* proteins are now termed CtNfs1, CtIsd11, CtIsu1, CtYfh1 and CtFdx2.

4. *The authors report that complexes lacking Acp have activity identical to complexes with added Acp. Have the authors rigorously determined the absence of Acp in complexes "lacking Acp". This needs to be done by MS analysis. Activity in the absence of Acp would seem to contradict the hypothesis that Acp is required and that acyl-Acp regulates activity.*

As also mentioned in our response to Reviewer 2 (section ACP), we now have added mass spectrometric data (new Fig. 4i).

The proposed regulatory function of ACP is not at all contradictory to our findings. The regulation would be via stabilization of the NFS1-ISD11 complex, as reported by Van Vrancken et al 2016. We have now better explained this in Discussion (paragraph "ACP is recruited...").

5. *A "regulatory" role for ISD11 is stressed over and over in the manuscript without much justification, aside from its not being "catalytic". Could it not also play a structural role, as indicated by studies showing that NFS1 is less stable in its absence?*

We do not think so. The Van Vrancken paper clearly suggests a stabilizing role of ACP binding to the NFS1-ISD11 high molecular mass complex. Together with our findings that ACP (like ISD11) in principle is dispensable for NFS1 desulfurase function, we think this clearly points to a regulatory role by NFS1 stabilization. We agree that further studies are needed to validate this conclusion.

6. *The rationale for using ISCU(M108I) rather than wild-type ISCU should be mentioned.*

We now have added a sentence to the Results and Methods to mention that this ISCU mutant protein was used in the expectation that frataxin would be dispensable as part of the core ISC complex (as demonstrated by Yoon, ..., Dancis 2012).

7. *In the ONLINE METHODS, it is unclear at times when NFS1 is being produced with a His-tag and when it is not.*

We have clarified this point. NFS1 contained His-tag only in constructs used for the crystallization of NIA complex. In all other cases NFS1 had no tag.

[1] Cory, S. A., Van Vranken, J. G., Brignole, E. J., Patra, S., Winge, D. R., Drennan, C. L., Rutter, J., and Barondeau, D. P. (2017) Structure of human Fe-S assembly subcomplex reveals unexpected cysteine desulfurase architecture and acyl-ACP-ISD11 interactions, *Proc. Natl. Acad. Sci. U. S. A.* 114, E5325-E5334.

This new paper is now discussed and cited, as mentioned in response to Reviewer 2.

Reviewer #4

..... and my comments will focus on the SAXS part. The SAXS data and their interpretation are convincing and support the claims of the author, however, a couple of points could be improved or specified:

- "Similar results were obtained by both species (Fig. 5a-b; supplementary Fig 5a-b and Table 2 and 3)". Plots and model of different species are on different figures and are therefore difficult to compare. The R_g of the different species reported in the table also differs between the species. It should be better explained why the results from the two species are similar.

Rephrased to "Similar envelopes were obtained for the human and *C. thermophilum* orthologs ..."

- In sup table 3 the R_g is given with significant digit up to the picometer. Rounding the R_g to the tenth of Angstrom like in sup table 2 (or even angstrom) would be more appropriate. The error should also be accordingly modified (together with the errors on the radius of gyration of (NFS1-FDX2)₂, (NFS1-FDX2-FXN)₂, (NFS1-ISCU-FDX2-FXN)₂ from sup table 2).

Done as suggested.

- For both species, the (NIA)₂ data were fitted and compared to crystallographic structure of (NIA)₂ alone and (NIA)₂ sub-complex taken from the (NIAU)₂ crystal. The agreement of the crystallographic structure with the SAXS data is illustrated using the *ab initio* shape and the χ^2 of the fit is given in the text. While *ab initio* model can provide the overall shape and are good for illustration, the fits of the SAXS curves computed from the crystallographic structure directly to the experimental data are much more convincing and could be shown in a figure.

A good suggestion. The fits are now incorporated into Fig. 5d and S5c. For more clarity, we also decided to incorporate the fits for the higher order complexes and added them to Fig. 6g and S6c.

- Along the same line, for the human (NIA)₂ (figure 5d), it would be much more convincing to underline directly the difference between the curves computed from the crystallographic structure and the experimental data rather than between the *ab initio* model and the crystal structure (with the red circle). (*Ab initio* models are build assuming a homogeneous electron density protein, which results in a loss of resolution.)

The calculated scattering amplitudes were added to Fig. S5d, top and compared to the experimental scattering curve of the (NIA)₂ complex. The fits of the two crystal structures to the envelope of the (NIA)₂ complex is now explicitly shown in Fig. S5d, bottom (Legend: "The bottom panel illustrates the fitting of the two complex parts into the SAXS envelope of (NIA)₂").

- It is not clear which *ab initio* shape are shown in the figures? From the methods section, one can understand that 20 models have been built but which one is represented, the most typical, the average, a refined shape?

The average shapes were used. We incorporated this into the Methods section.

- In the part solution structures of NFS1, ISCU, FDX2 and FXN sub-complexes, it is not clear from the text if all the models of the complexes are obtained by rigid body modelling (with sasref). (+ If this is the case, and if the positions of the different monomer are the same in the

different complexes, a rigid body approach where the SAXS curves of the different complex would be simultaneously fitted and the models of the different complex simultaneously built (imposing known interfaces between monomers if needed) could lead to more robust models.)

Yes, all models were calculated by sasref. Method section: “For final rigid body modelling of all proteins/complexes, SASREF was used”. This applies to all models shown.

We tried a rigid body modeling approach using the different scattering curves and sub-complexes (or monomers). It turned out (especially in the case of NFS1-FXN and NFS1-FDX2) to be impossible to define the exact interface residues due to the lack of high resolution structural data of these complexes. Therefore, we were not successful to produce a reliable “contact conditions file”. We therefore decided to assemble the complexes stepwise, and performed rigid body modeling with the respective refined docking models for each scattering data separately.